


# Global CO₂ emissions from cement production

Robbie M. Andrew[1]

[1]CICERO Center for International Climate Research, Oslo 0349, Norway

*Correspondence to*: Robbie M. Andrew (rm.andrew.nz@gmail.com)

**Abstract** Global production of cement has grown very rapidly in recent years, and after fossil fuels and land-use change, it is the third-largest source of anthropogenic emissions of carbon dioxide. The required data for estimating emissions from global cement production are poor, and it has been recognised that some global estimates are significantly inflated. Here we assemble a large variety of available datasets, prioritising official data and emission factors, including estimates submitted to the UNFCCC plus new estimates for China and India, to present a new analysis of global process emissions from cement production. We show that global process emissions in 2017 were $1.48 \pm 0.20$ Gt $CO_2$, equivalent to about 4% of emissions from fossil fuels. Cumulative emissions from 1928 to 2017 were $36.9 \pm 2.3$ Gt $CO_2$, 70% of which have occurred since 1990. Emissions in 2016 were 28% lower than those recently reported by the Global Carbon Project. The data associated with this article can be found at https://doi.org/10.5281/zenodo.831454.

## 1 Introduction

Anthropogenic emissions of carbon dioxide to the atmosphere come from three main sources: (i) oxidation of fossil fuels, (ii) deforestation and other land-use changes, and (iii) carbonate decomposition. Cement – the largest source of emissions from decomposition of carbonates – is a binding material that has been used since ancient times. But it was following World War II that the production of cement accelerated rapidly worldwide, with current levels of global production equivalent to more than half a tonne per person per year (Figure 1). Global cement production has increased more than 30-fold since 1950, and almost four-fold since 1990, with much more rapid growth than global fossil energy production in the last two decades. Since 1990 this growth is largely because of rapid development in China, where cement production has grown by a factor of more than 11, such that 75% of global growth in cement production since 1990 occurred in China.

There are two aspects of cement production that result in emissions of $CO_2$. First is the chemical reaction involved in the production of the main component of cement, clinker, as carbonates (largely $CaCO_3$, found in limestone) are decomposed into oxides (largely lime, $CaO$) and $CO_2$ by the addition of heat. Stoichiometry directly indicates how much $CO_2$ is released for a given amount of $CaO$ produced. Recent estimates are that these so-called 'process' emissions contribute about 5% of total anthropogenic $CO_2$ emissions excluding land-use change (Boden et al., 2017). The second source of emissions is from the combustion of fossil fuels to generate the significant energy required to heat the raw ingredients to well over 1000°C, and these



'energy' emissions, including those from purchased electricity, could add a further 60% on top of the process emissions (IEA, 2016). Total emissions from the cement industry could therefore contribute as much as 8% of global $CO_2$ emissions. These process (sometimes 'industry' or 'industrial process') and energy emissions are most often reported separately in global emissions inventories (Eggleston et al., 2006; Le Quéré et al., 2018).

5   The Global Carbon Project annually publishes estimates of global emissions of $CO_2$ from use of fossil fuels and cement production, and these estimates are used by the global carbon modelling community as part of development of the Global Carbon Budget (Le Quéré et al., 2018). It is therefore important that the emissions estimates are as accurate as possible. This emissions database covers all emissions of $CO_2$ resulting from oxidation (not only energy-use) of fossil fuels, including those that occur in the IPCC sector 'Industrial Processes and Product Use', such that including cement emissions means that the vast

10   majority of $CO_2$ emissions are covered.

In this work we investigate the process emissions from cement production and develop a new time series for potential use by the Global Carbon Project, and present plans for future continued updates, revisions and development. The focus on process emissions here is because both direct fossil fuel emissions and electricity emissions are already accounted for in other parts of the Global Carbon Budget.

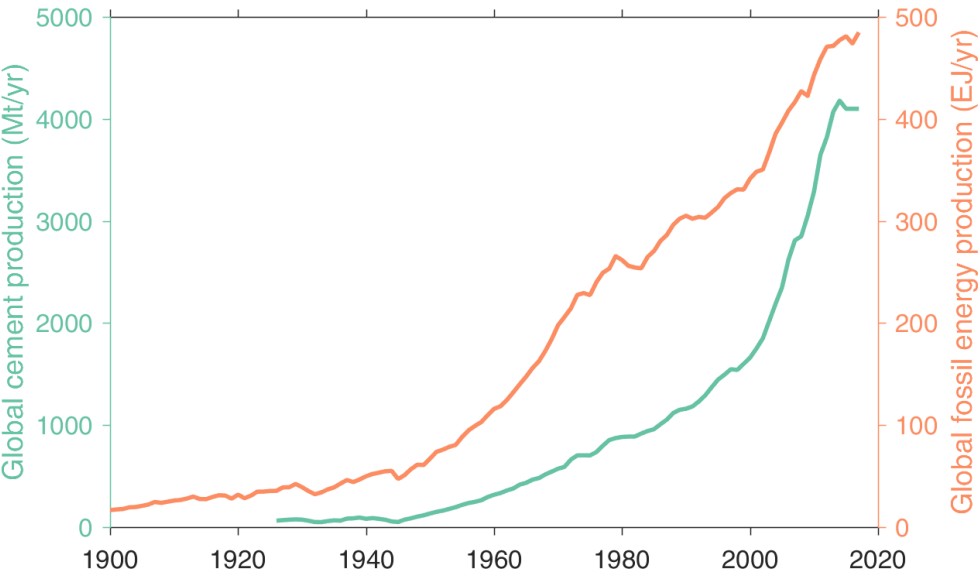

**Figure 1: Global cement and fossil energy production to 2017 (USGS, 2014, 2018; Mohr et al., 2015; BP, 2018).**

**2 Previous Estimates of Global Cement Emissions**

Early estimates of emissions from global cement production effectively assumed that almost all cement was of the Ordinary Portland Cement (OPC) type, which uses a very high proportion of clinker and very small amounts of other ingredients, such



as gypsum to control setting time. For at least the first half of the 20th Century this assumption was quite reasonable, with the vast majority of cement being produced in industrialised countries, which followed carefully developed and tested standards regarding strength and other important qualities.

In 1970, Baxter and Walton presented estimates of global $CO_2$ emissions from fossil fuels and cement production for 1860– 1969, where the "mean calcium oxide content of cements was taken to be 60 % … and the carbon content of limestone assumed to be 12% with 100% kilning efficiency (Baxter and Walton, 1970). Thus the …. manufacture of 1 tonne of cement yields … 4.71 x $10^5$g of carbon dioxide…" (i.e. 0.471 t $CO_2$ (t cement)$^{-1}$. Assuming their estimate of global cement production in 1969 was the same as that reported by the USGS (2014), their estimate of emissions from cement production in 1969 would have been 256 Mt $CO_2$.

In a landmark paper of 1973, Charles Keeling presented a systematic analysis of emissions from fossil fuel combustion for 1860–1969 and cement production for 1949–1969 (Keeling, 1973). Using an average CaO content of cement of 64.1%, Keeling's emission factor was 0.50 t $CO_2$ (t cement)$^{-1}$, giving an estimate for emissions from cement production in 1969 of 272 Mt. While both Keeling (1973) and Baxter and Walton (1970) cited Lea and Desch (1940) as the source for their estimates of the CaO content of cement, they nevertheless used different fractions. Importantly, these fractions were assumed to be time-invariant.

Marland and Rotty (1984) presented further estimates for 1950–1982, using a global average CaO content of cement of 63.8%, taken directly from US data for 1975. From this they derived a time-invariant emission factor of 0.50 t $CO_2$ (t cement)$^{-1}$.

The estimates made by Marland and Rotty (1984), combined with the earlier estimates of Keeling (1973) were included in the archive of the Carbon Dioxide Information Analysis Center (CDIAC) in 1984 (Rotty and Marland, 1984). Later, CDIAC modified the cement emission factor very slightly based on a study by Griffin (1987), who (in turn based on Orchard (1973)) said that "the range of lime [CaO] content in cement is 60–67 percent" and, based on discussion with experts, recommended the use of 63.5%, calculated as the midpoint of the range (Boden et al., 1995). This time-invariant, global emission factor of about 0.50 was still in use in CDIAC's 2016 data release.

CDIAC's method was directly adopted by the Intergovernmental Panel on Climate Change (IPCC) in their 1996 guidelines (Haukås et al., 1997) in the case where clinker production data were not available. The IPCC subsequently revised its methods in the case where clinker production are not available, in the 2006 Guidelines (p2.8):

> *[I]n the absence of data on carbonate inputs or national clinker production data, cement production data may be used to estimate clinker production by taking into account the amounts and types of cement produced and their clinker contents and including a correction for clinker imports and exports. Accounting for imports and exports of clinker is an important factor in the estimation of emissions from this source.*

In addition, the IPCC Guidelines now recommend use of a default clinker ratio of 0.75 when it is known that significant amounts of blended cements are produced.

The Emissions Database for Global Atmospheric Research (EDGAR) presents estimates of $CO_2$ and other climate-important gases by country. For cement they initially used the emission factor from Marland and Rotty (1984) of 0.50 t $CO_2$ (t cement)$^{-}$



[1] (Olivier et al., 1999). With the release of version 4.1 of the database in 2010, they modified their emission factor to account for changing rates of blending (i.e., lower clinker ratios) in cement production in response to work by the World Business Council for Sustainable Development (WBCSD), who released sample-based estimates of the clinker ratio in a range of countries (Anonymous, 2010). In version 4.3.2, EDGAR used official estimates from Annex-I Parties to the UNFCCC, specific clinker production data for China, and the WBCSD database for all remaining countries (Olivier et al., 2016; Janssens-Maenhout et al., 2017).

Since 2003, countries that are listed in Annex 1 of the UN Framework Convention on Climate Change (UNFCCC) have submitted annual inventories of greenhouse gas emissions in considerable detail, including estimates of emissions from cement production (UNFCCC, 2018). Other Parties to the Convention are requested to submit less detailed and less frequent National Communications and, more recently, Biennial Update Reports (BURs).

## 3 Methods

While cement production data are available by country (van Oss, 1994–2018), it is production of clinker that leads to process $CO_2$ emissions, and the amount of clinker in cement varies widely. With no available source of clinker production data for all countries, other options must be considered. The direct use of cement production data without adjustment for clinker ratios that vary by country and over time, or for clinker trade, leads to poor emissions estimates (see Appendix A), and should therefore be used only as a last resort. The World Business Council for Sustainable Development (WBCSD), through its 'Getting the Numbers Right' initiative, has collected cement data, including clinker production data, directly from firms, but their survey-based approach leaves many parts of the world poorly sampled (WBCSD, 2014).

The main rationale of our approach, therefore, is to prioritise officially reported emissions, recognising that these generally make use of data and knowledge unavailable elsewhere; then we use officially reported clinker production data and emission factors; then IPCC default emission factors; then industry-reported clinker production; and finally survey-based clinker ratios applied to cement production data, where no better data are available. Full details are provided in Appendix D and in the associated data files.

For the 42 Annex-I countries that report their greenhouse gas inventories annually to the UNFCCC, we extract official estimates of cement-production emissions from 1990 onwards. Some eastern-European countries submit data for years before 1990: Poland and Bulgaria from 1988, Hungary from 1986, and Slovenia from 1987. These are all based on clinker production data and largely use Tier-2 methods. This dataset covers about 10% of current global cement production, and is available as consistently structured spreadsheet files for each year. In addition, clinker production data were available for the US from 1925 (Hendrik van Oss, USGS, personal communication 2015).

Some non-Annex-I Parties have begun to include time-series of cement emissions in their National Communications, National Inventory Reports, and Biennial Update Reports to the UNFCCC, and these estimates have been used directly. At the time of writing, the following countries reported useable time-series data: Armenia, Azerbaijan, Bosnia and Herzegovina, Brazil,





Chile, Colombia, Indonesia, Israel, Jamaica, Lebanon, Mexico, Moldova, Mongolia, Morocco, Namibia, Serbia, South Africa, Togo, and Uzbekistan. In addition, Brunei Darussalam. Côte d'Ivoire, Mauritania, Sierra Leone, and Tuvalu report that all of their clinker is imported.

For China, which currently produces almost 60% of global cement, clinker production data is available from 1990. China's emission factor is reported by NDRC (2014) as 0.5383 t $CO_2$ (t clinker)$^{-1}$, and this is used both in the Second National Communication (NDRC, 2012) and the First Biennial Update Report (NDRC, 2016). Some studies have estimated other emission factors based on factory-level sampling (Liu et al., 2015; Shen et al., 2014), but here we use the officially sanctioned factor until or unless that is changed.

India, the world's second-largest cement producer with about 7% of global production in recent years, does not officially report clinker production statistics. Data from the Cement Manufacturers' Association (CMA) are useful only until the 2009/10 financial year, when two large producers discontinued membership of the organisation (CMA, 2010). Clinker production data are also reported by business consultancies in their annual overviews of the industry in India. Data on the types of cement produced, combined with their likely clinker contents, can also be used to support this evidence base.

While Jamaica reported cement emissions for 2006–12, the data source was clearly identified and additional clinker production data has been obtained to cover 1995–2016. Meanwhile, clinker production data for the Republic of Korea were readily available from its Cement Association for 1991–2016; emissions estimates from these data matched those reported in official communications to the UNFCCC during overlapping periods. Clinker production data were also available for Saudi Arabia from one of its cement manufacturers for 2003–2017.

Finally, for all remaining countries we have used survey-based clinker-ratio data from the WBCSD's Getting the Numbers Right initiative (WBCSD, 2014), combined with historical cement production data from the USGS. In many cases these clinker ratios are presented only for groups of countries, but nevertheless represent the best available information about clinker ratios in those countries.

Most of these methods provide estimates only back to 1990 at best, and we therefore extrapolate for earlier years using cement production data combined with assumptions about how clinker ratios have changed over time. We make the basic assumption that most countries began their cement industries by producing Ordinary Portland Cement, a strong and very common cement type with a clinker ratio of 0.95, and over time introduced other types of cements with lower clinker ratios. This assumption reflects available observations. Specifically, the clinker ratio was set to 0.95 in 1970, with the IPCC default emission factor, and linearly interpolated to the implied ratio and emission factor in the earliest year for which data are available for each country. For large cement producers, covering more than 80% of global production, USGS provides an estimate of cement production for 2017 (USGS, 2018), and these are used to estimate 2017 emissions for those countries. For other countries emissions are assumed to be the same as in 2016. While this extrapolation is clearly not ideal, not extrapolating would result in very large discontinuities and frustrate any attempt at trend analysis, and particularly any assessment of cumulative emissions. Extrapolating necessarily affects derived growth rates, but these growth rates are dominated by the changes in cement production much more than the extrapolation method.



It is clear from this that data quality is significantly higher from 1990 onwards, and estimates before then will have higher uncertainty. However, emissions prior to 1990 are also less important in the global policy debate, and, because only about 30% of historical cement production occurred before 1990, emissions from that period are of lower importance also for global carbon modelling and budget calculations. In addition, the rate of change of technology was much slower before 1990, with

5    most adjustments to, for example, the clinker content of cement, occurring in more recent times, so that estimates for earlier years are less sensitive to assumptions. We estimate uncertainty in global cement emissions using a Monte Carlo approach, as described in Appendix C.

## 4 Results

Process emissions from cement production reached a new peak in 2017 of $1.48\pm0.20$ $GtCO_2$, after declining the previous two

10    years (Figure 2). In comparison, CDIAC's estimate for 2014 is 2.08 $GtCO_2$ (Boden et al., 2017). The most recent estimate currently available from EDGAR is for 2015, at 1.44 $GtCO_2$ (Olivier et al., 2016), in very good agreement with our estimate for the same year of $1.47\pm0.11$ $GtCO_2$. Cumulative emissions over 1928–2017 were $36.9\pm2.3$ $GtCO_2$. The global-average clinker ratio has declined from approximately 0.83 in 1990 to 0.67 in 2013 (Figure E1) – consistent with an estimate of 0.65 made by the IEA (IEA, 2017) – before rebounding slightly to 0.69 in 2017.

For China, emissions reached just under 800 $MtCO_2$ in 2014 (Figure 3). The emissions estimated here show high agreement with the few official estimates reported, a direct consequence of our use of official data and emission factors. While China produced 57% of the world's cement in 2017, its emissions were 52% of the total, a consequence of its clinker ratio being less than 0.60 in recent years, below the world average. Results for a number of other countries are presented in the Appendices.

Indian emissions are quite uncertain, but the methods used here produce results reasonably close to the few officially reported

estimates (Figure 4). In 2010 there is some divergence from the estimate in India's first Biennial Update Report. In that year the data provided by the Indian Cement Manufacturers' Association are known to be incomplete, while other data sources indicate substantially higher clinker production in that year; this discrepancy is yet to be resolved (see Appendix D).

Aggregate uncertainty is relatively low through most of the historical period (Figure 2, top panel), partly as a direct consequence of the choice of the Monte Carlo method with symmetric distributions and no correlation: errors tend to cancel.

In 1990, with the beginning of most Annex-I countries' detailed reporting to the UNFCCC, global uncertainty declines slightly, but then gradually increases as more cement production occurs in developing countries, where uncertainty is higher. Uncertainty jumps in 2017 because of the use of more provisional data.

A recent study estimated global cement carbonation (uptake of $CO_2$ by concrete during its use and disposal phases) at about 900 $MtCO_2$ in 2013 (Xi et al., 2016), which would be about 63% of emissions from cement production in that year. However,

the central estimate (within a Monte Carlo uncertainty assessment) was based on the assumption that the global average clinker ratio was 0.75, the default suggested by the IPCC for countries with a significant proportion of blended cement production (Hanle et al., 2006). Interestingly, while the global clinker ratio appears to be substantially lower than 0.75, the important





scaling factor in the estimate of carbonation is in fact the CaO content of the concrete, and use of clinker substitutes means that global carbonation could actually be higher rather than lower than the central estimate of Xi et al. (2016).

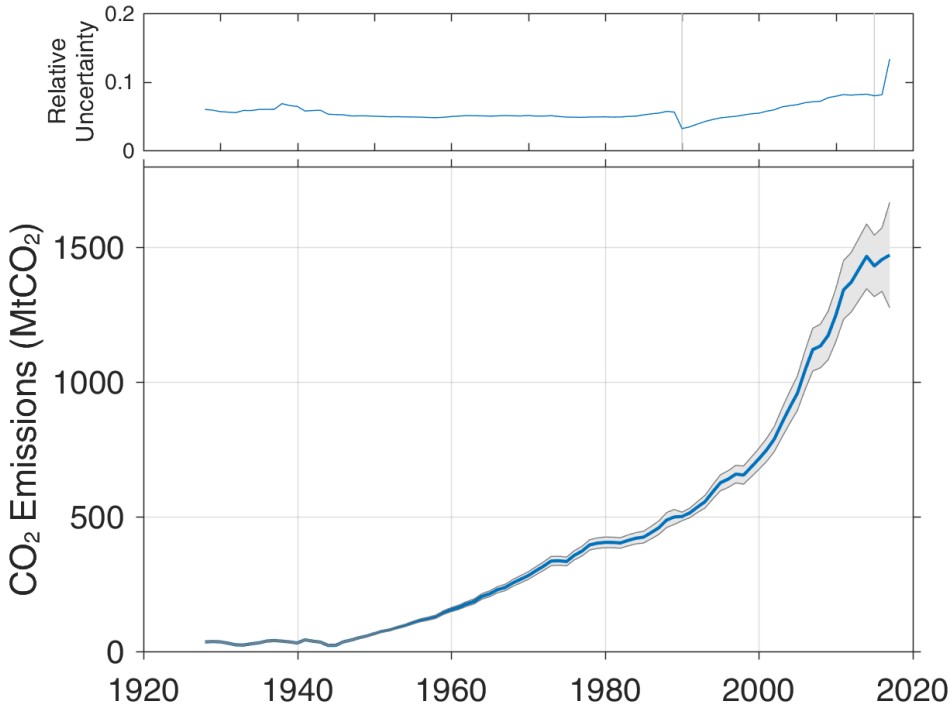

**Figure 2: Global process emissions from cement production, with 95% confidence interval. A step change in uncertainty occurs in 1990, reflecting a significant change in data availability.**




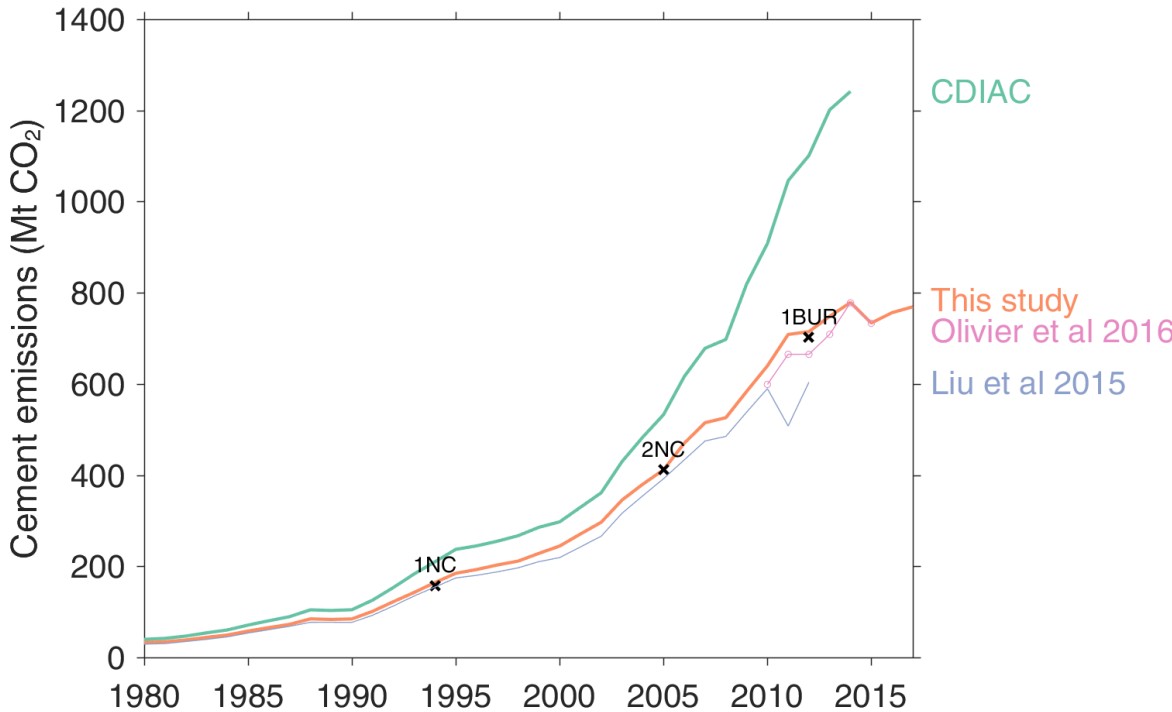

**Figure 3: Process emissions from Chinese cement production, 1980–2017. 1NC refers to China's First National Communication, 2NC the Second, and 1BUR the first Biennial Update Report. Also shown are estimates from CDIAC (Boden et al., 2017), Liu et al. (2015) and EDGAR v4.3.2 FT2015 .**





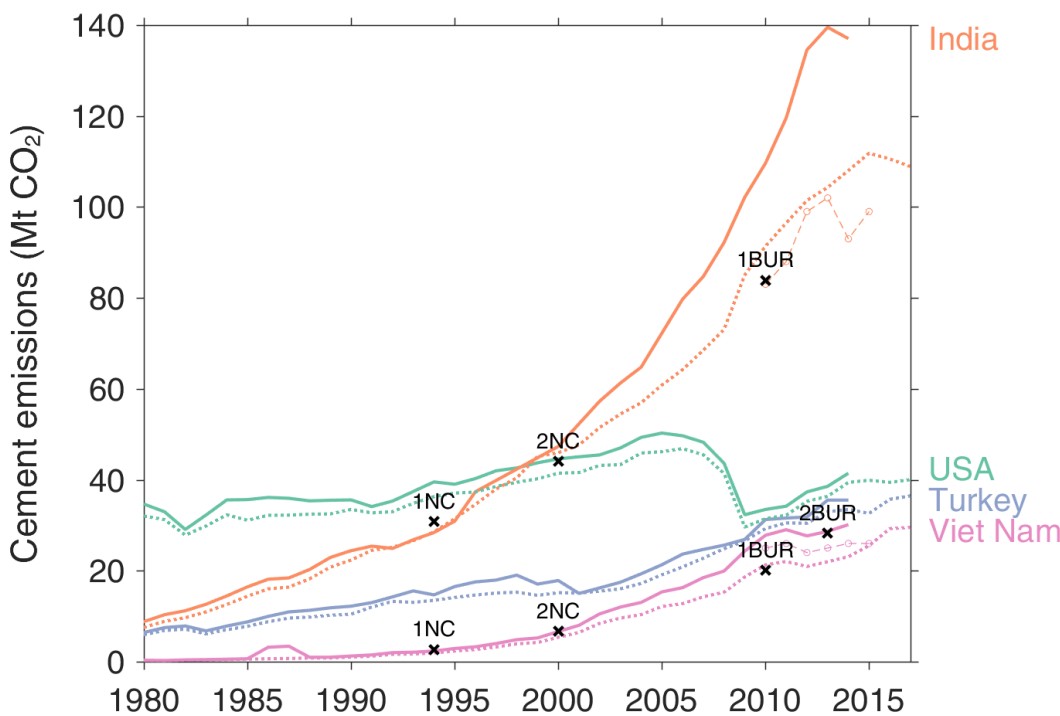

**Figure 4: Comparing new cement emissions estimates (dashed lines) for the top four cement producers after China with those from CDIAC (solid lines), and official estimates (crosses, India and Viet Nam) as reported to the UNFCCC (see text). The new estimates for the USA and Turkey come directly from national official estimates. Estimates from EDGAR v4.3.2_FT2015 are shown for India and Viet Nam with round markers.**

## 6 Data Availability

All data used in producing this dataset, and the resulting dataset itself, are available on Zenodo at the following DOI: https://doi.org/10.5281/zenodo.831454.

The exception is the "Getting the Numbers Right" dataset from WBCSD, which is available from their website: http://www.wbcsdcement.org/GNR-2014/index.html.

## 7 Conclusions

Estimating global process emissions from cement production is fraught with problems of data availability, and has always required strong assumptions. Over the last three decades, countries around the world have increasingly been producing blended cements, with lower clinker ratios, and the use of cement production data with constant emission factors has become untenable.



The new global cement emissions database presented here increases the reliance on official and reliable data sources, and reduces reliance on assumptions, compared with previous efforts. It is intended that the database will be used in the Global Carbon Budget and updated annually, with both data updates and methodological improvements. As more countries estimate their emissions and report them to the UNFCCC in detail, more data will replace assumptions in producing this dataset. Work is still required in improving estimates of cement emissions from both China and India, in particular, as these are the world's two largest cement producers and official time-series estimates are lacking.

**Appendix A: Reasons for different estimates**

Released annually, CDIAC's emissions estimates are widely reported, including in the IPCC's Fifth Assessment Report (Ciais et al., 2013). However, recently there have been some questions raised about the accuracy of CDIAC's cement emissions estimates, particularly for China (e.g., Lei, 2012; Ke et al., 2013; Liu et al., 2015). According to Ke et al. (2013), CDIAC's estimates of cement emissions for China were 36% higher than those obtained from an IPCC Tier 2 method for 2007, amounting to an 'error' of 181 $MtCO_2$, noting that "CDIAC's relatively higher emission factor is equivalent to the assumption of a high clinker-to-cement ratio" (p. 175).

**A1 Clinker ratios**

The most obvious reason that CDIAC's estimates are higher than those produced elsewhere is that the formula they have used obscures an assumption about the ratio of clinker to cement in production.

CDIAC's method for estimating process emissions from cement production by country is taken from a report by Griffin (1987), and requires that cement production data in tonnes are multiplied by a fixed factor 0.136 to obtain tonnes of carbon emitted as $CO_2$, i.e., 1 tonne of cement produced results in $0.136 \times 3.667 = 0.50$ t $CO_2$ (Boden et al., 1995).

According to Griffin (1987), the emissions factor for the production of cement, $E_{cem}$, from the calcination of limestone is given as:

$$E_{cem} = f_{cem}^{CaO} \frac{M_r^{CO_2}}{M_r^{CaO}}$$

where $f_{cem}^{CaO}$ is the fraction of CaO in cement, $M_r^{CO_2}$ is the molecular weight of $CO_2$ (44.01), and $M_r^{CaO}$ is the molecular weight of CaO (56.08). Based on discussion with experts, Griffin (1987) recommended that $f_{cem}^{CaO} = 0.635$, calculated as the midpoint of the range 0.60–0.67 given by Orchard (1973).

According to the IPCC's more recent 2006 Guidelines (Hanle et al., 2006), when using cement production data adjusted for clinker trade, the formula should read:

$$E_{cem} = f_{cem}^{clink} f_{clink}^{CaO} \frac{M_r^{CO_2}}{M_r^{CaO}}$$



where $f_{cem}^{clink}$ is the clinker ratio, and $f_{clink}^{CaO}$ is the fraction of CaO in clinker. In the earlier, 1996 IPCC Guidelines, the information sourced from CDIAC stated that the average CaO content of cement is 0.635, while the CaO content of clinker is 0.646, yielding an implicit average clinker ratio of cement of 0.98.

This high implicit clinker ratio appears to be based on the assumption that the majority of cement produced in the world is

(was) Ordinary Portland cement: "Other speciality cements are lower in lime, but are typically used in small quantities. … The differences between the lime content and production of clinker and cement, *in most countries*, are not significant enough to affect the emission estimates" (Haukås et al., 1997, p2.5; emphasis in original). Indeed, Orchard (1973) made his statement about lime content in reference to Portland cements, which are that type that is composed of at least 95% clinker, rather than cement in general.

In the USA, the average clinker ratio was most likely about 0.95 for much of the 20[th] century, possibly dropping to about 0.90 or slightly lower after about 1970 (Hendrik van Oss, USGS, personal communication, 2015). However, the International Energy Agency (IEA), recently estimated the global-average clinker ratio to be 0.65 (IEA, 2017), and the dataset presented in this work agrees with that assessment (Appendix E). In China, where almost 60% of cement is produced, the clinker ratio is currently below 0.60.

WBCSD demonstrate that the clinker ratio has been declining in every region, and, based on the data they have available, the world average for 2012 was about 0.75. Furthermore, between 2000 and 2006 the clinker ratio decreased more quickly in developing countries than developed countries. WBCSD puts the primary reason for a lack of decline in developed countries as the acceptance of common practice and fixed product standards, which act as a barrier to reduction in clinker content. This is in contrast to, in particular, India and China, where fly ash from coal-fired power stations and slag from the iron and steel

industry are widely used as clinker substitutes (WBCSD, 2009). Interestingly, it may simply be more common practice in developed countries for the construction industry to blend in other ingredients after the cement is made but before its use (AT Kearney, 2014).

## 2. Use of cement production data

The best available data on $CO_2$ emissions from cement production at a national level come from official submissions to the

UNFCCC, with about 40 countries submitting annually (UNFCCC, 2017). Figure A1 compares $CO_2$ emissions from CDIAC with those from UNFCCC specifically for the process of calcination. Over the 26-year period covered by the UNFCCC submissions (1990–2015), CDIAC's estimates are on average 11% higher than those estimated by these countries. All countries reporting to the UNFCCC use clinker production data to estimate $CO_2$ emissions.



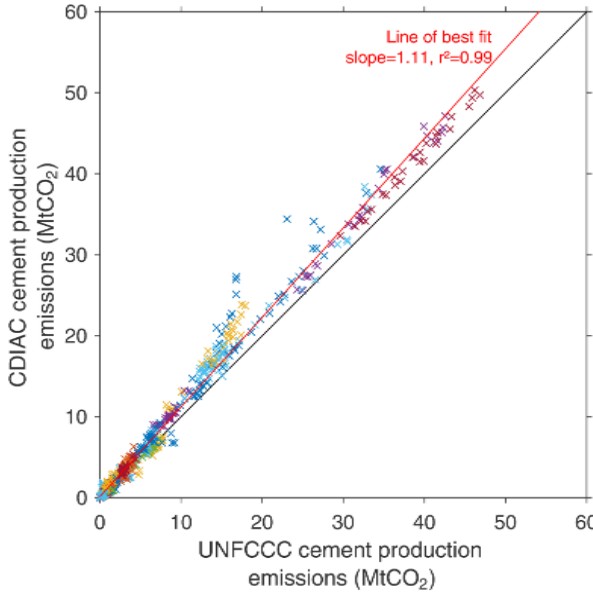

**Figure A1: Comparison of CO$_2$ emissions in 43 countries as estimated by CDIAC (Boden et al., 2013) and those officially reported to the UNFCCC, 1990-2012 (UNFCCC, 2014).**

CDIAC's estimates are produced using cement production data obtained from the USGS. However, according to the IPCC

5 Guidelines (Hanle et al., 2006, p2.8),

> "[C]alculating CO$_2$ emissions directly from cement production (i.e., using a fixed cement-based emission factor) is not consistent with good practice. Instead, in the absence of data on carbonate inputs or national clinker production data, cement production data may be used to estimate clinker production by taking into account the amounts and types of cement produced and their clinker contents and including a correction for clinker imports and exports. Accounting for
>
> 10 imports and exports of clinker is an important factor in the estimation of emissions from this source."

There is clearly some noise around the line of best-fit comparing CDIAC's estimates to emissions reported to the UNFCCC, as shown in Figure A1, such that simply adjusting estimates down by 11% (implying an average clinker ratio of about 0.87 for these countries) would still leave considerable differences with official estimates for some countries. These deviations could be explained as the effects of varying clinker ratios and international trade of clinker. The more clinker is imported for cement

15 production (or exported), the poorer cement production data become for the purpose of estimating cement emissions.

The Netherlands provides a clear example of how poor the use of cement production data and a global-average clinker ratio can be. CDIAC's emissions estimates are at least double those reported to the UNFCCC, and as much as four times as high (Figure A2: left). The reason for this is because of significant net imports of clinker and a particularly low clinker ratio (Figure A2: right). The low clinker ratio is because most of the country's production is of cement type CEMIII, which is specifically

20 suitable for use in marine conditions (CEMBUREAU, 2013), and this type of cement uses a much lower clinker ratio (European standard 197-1).



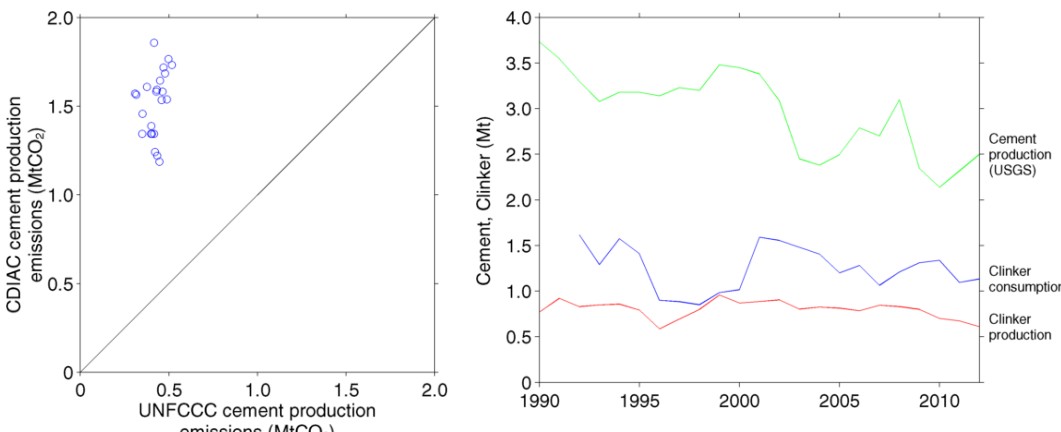

**Figure A2: Netherlands. Left: CDIAC vs UNFCCC. Right: Clinker, cement. Note 'Clinker consumption' is production plus imports less exports, but excludes stock changes. Sources: (UNSD, 2015; UNFCCC, 2014; van Oss, 1994–2018; Boden et al., 2013).**

## 3. System boundaries

As has been identified by others, one of the reasons for divergences between estimates of cement emissions is that different system boundaries have been used (e.g., Shen et al., 2014; Ke et al., 2013). Studies vary on whether they include process emissions from clinker production, other process emissions, direct fuel combustion emissions, and emissions from generation of purchased electricity. The IPCC Guidelines clearly delineate types of emissions, and process emissions from clinker production are allocated to the Industrial Processes and Product Use (IPPU) sector, while emissions from electricity generation or direct fuel combustion by clinker producing firms are allocated to the Energy 'sector' (Eggleston et al., 2006). Sometimes lime is produced and mixed with clinker, and emissions from this process are also allocated to the IPPU sector, but listed separately from cement emissions.

It is not widely understood that CDIAC's emissions estimates do not follow the IPCC delineations, and instead CDIAC estimates emissions resulting from all oxidation of fossil fuels plus those from cement production (Boden et al., 1995; Marland and Rotty, 1984; Andres et al., 2012). Therefore, CDIAC's estimates of emissions from coal oxidation include non-energy use of coal, such as when used for anodes in Aluminium production, in contrast to the IPCC methodology. CDIAC's system boundary is therefore much broader than generally understood, including as it does not only all energy emissions but also most industrial process emissions.

## Appendix B: Cement production data

In this work, historical cement production data in tonnes are sourced from CDIAC's cement emissions data. Because CDIAC use a constant emission factor based on cement production, reverse-calculation of cement production data is straightforward. Those production data came originally from USGS (formerly Bureau of Mines; Marland and Rotty, 1984). This is significantly less time-consuming than replicating CDIAC's work of assembling USGS's various datasets.



## Appendix C: Uncertainty analysis

Our uncertainty analysis leans heavily on the officially estimated uncertainty of cement emissions provided in submissions to the UNFCCC, whether in National Inventory Reports, National Communications, or Biennial Update Reports. These uncertainties, which follow the methods outlined in the IPCC's guidelines (Eggleston et al., 2006), represent 2 SD of a normal

distribution (95%). For countries without official estimates of uncertainty, estimates have been made based on the approaches used and other information. The greatest uncertainty is when only cement production data and average clinker ratios have been used, and for these cases the uncertainty (2 SD) has been set at 25%. See the accompanying uncertainty dataset for details.

We have also allowed uncertainty to vary by time, with much higher uncertainties outside of the time covered by official estimates. For example, Annex-I countries report emissions for 1990–2016, while outside of that period clinker ratios and

10 cement production data have been used, with higher uncertainty.

The uncertainty estimates by country and by time are used in a Monte Carlo analysis with 10,000 runs to give estimates of uncertainty for global cement emissions. This method effectively uses combined uncertainty of all underlying factors, such as method, clinker ratios, emission factors, cement kiln dust factors, and so on.

Uncertainties are assumed to be uncorrelated between countries and across time. The later assumption means that the

15 uncertainty of any derived growth rates would be overestimated.

The results of the uncertainty analysis at the global level are shown in the main text, Figure 2.

## Appendix D: Country-specific analyses

### D1 Annex I Parties to the UNFCCC

The following countries report annual emissions inventories to the UNFCCC using the Common Reporting Framework (CRF),

and these were downloaded on 25 April 2018, except for Ukraine whose data were downloaded on 24 May 2018. UNFCCC Parties sometimes submit revisions through the year, and the specific date of each country's submission as used in this study is shown here:

Australia: 13/04/2018, Austria: 10/04/2018, Belgium: 13/04/2018, Bulgaria: 13/04/2018, Belarus: 06/04/2018, Canada: 09/04/2018, Switzerland: 16/04/2018, Cyprus: 11/04/2018, Czech Republic: 06/04/2018, Germany:

04/04/2018, : 14/04/2018, Denmark: 14/04/2018, Spain: 02/04/2018, Estonia: 13/04/2018, Finland: 06/04/2018, France: 12/04/2018, United Kingdom: 16/04/2018, Greece: 04/04/2018, Croatia: 27/03/2018, Hungary: 13/04/2018, Ireland: 12/04/2018, Iceland: 12/04/2018, Italy: 12/04/2018, Japan: 18/04/2018, Kazakhstan: 23/04/2018, Liechtenstein: 09/04/2018, Lithuania: 13/04/2018, Luxembourg: 04/04/2018, Latvia: 12/04/2018, Malta: 11/04/2018, Netherlands: 10/04/2018, Norway: 13/04/2018, New Zealand: 10/04/2018, Poland: 09/04/2018, Portugal:

02/04/2018, Romania: 16/04/2018, Russia: 13/04/2018, Slovakia: 26/03/2018, Slovenia: 13/04/2018, Sweden: 11/04/2018, Turkey: 13/04/2018, Ukraine: 22/05/2018, United States of America: 12/04/2018.



These inventories explicitly state process emissions from cement production from 1990 onwards (IPCC sector 2A1). The 2018 submissions include emissions data up to 2016. Monaco's emissions have been combined with those of France, following CDIAC.

Figure D1 compares cement emissions for Annex-I Parties as reported by CDIAC (Boden et al., 2017) with those reported here[1].

---

[1] Note that in all the figures that follow, 'Official' indicates the use of either officially reported emissions estimates or official/semi-official national clinker production estimates. In each case the text explains the sources used.





**Figure D1: Revised cement emissions for Annex-I parties to the UNFCCC.**





**Figure D1 (cont.): Revised cement emissions for Annex-I parties to the UNFCCC.**





**Figure D1 (cont.): Revised cement emissions for Annex-I parties to the UNFCCC.**



## D2 China

As by far the largest producer of cement worldwide, estimating China's emissions from cement production is critical to having a robust global estimate. In 1982 China overtook Japan to become the world's largest producer of cement and in 2017 accounted for about 57% of global production (Figure D2).

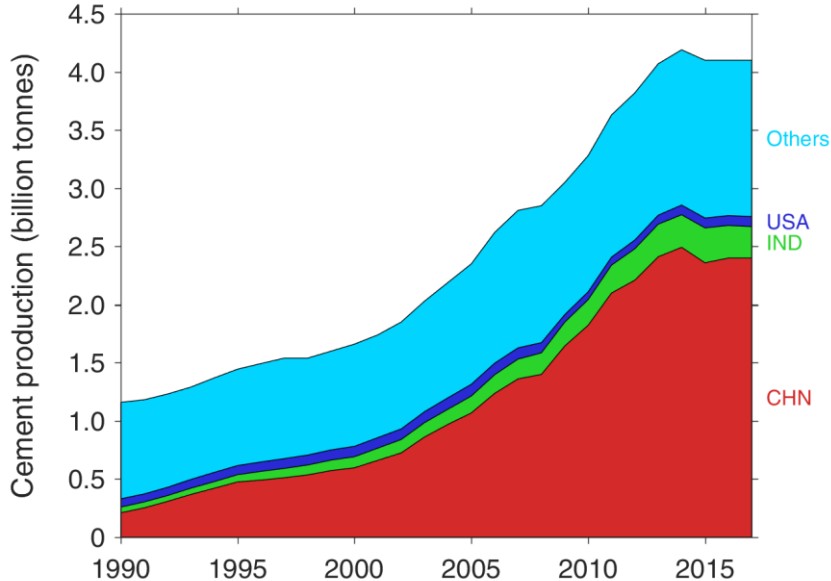

**Figure D2: Production of cement by country, 1990–2017 (van Oss, 1994–2018; USGS, 2018).**

China has released several official estimates of process emissions from cement production in reporting to the UNFCCC. In its First National Communication to the UNFCCC, China reported[2] process emissions from cement production of 157.8 Mt CO2 in 1994 from about 300 Mt clinker (SDPC, 2004). In its Second National Communication, China reported[3] 411.7 Mt $CO_2$ in 2005 from about 765 Mt[4] of clinker (NDRC, 2012, 2014). And in its first Biennial Update Report, China doesn't report emissions from cement production separately, but does report[5] clinker production of 1303.9 Mt in 2012 (NDRC, 2016), which,

---

[2] Page 32.

[3] Page 59.

[4] Page 39 of the Second National Communication actually says 674, but this is a typographic error. The NDRC's 2005 GHG Inventory Research book gives 764.71 Mt clinker production in 2005 NDRC: The People's Republic of China National Greenhouse Gas Inventory 2005, National Development and Reform Commission, Beijing, 2014., which agrees both with the figure given by CCA – 764.72 Mt – and with the reported emissions.

[5] Table 2-3, on page 20 in the English section [p152].





with China's emission factor of 0.5383, would have led to about 702 MtCO$_2$. In all three cases, China has used firm-level surveys to determine the emission factor.

In 2016 the China Cement Association's (CCA) annual Cement Almanac 2015 presented much lower historical clinker production for some years than previous editions (CCA, 2016 (in Chinese)). These are not revisions, but a change in the coverage of the data presented: previous Almanacs presented national totals, while the 2015 edition presents production from enterprises with revenues over a specified threshold (so-called "above-sized" enterprises; a correspondent at CCA, personal communication, 2017). The differences between these two figures has diminished considerably over time, such that clinker production from above-size enterprises in 2013 was 98% of all clinker production reported by CCA in the previous edition.

National clinker production data for 1990–2004 were provided by Shaohui Zhang, who received them directly from CCA (Zhang et al., 2015); 2005–2013 are from the 2015 edition of CCA's Almanac; 2014–2017 are from NBS via the China Cement Research Institute (CCRI), and these have been scaled up very slightly so that the 2013 figure matches the national total provided by CCA.

Figure D3 shows clinker ratios (the ratio of clinker production to cement production) from this and a number of other sources. Some authors do not adjust for clinker trade before calculating the ratio. The numbers from WBCSD are unreliable because of a very small sample size in China (~4% of all clinker production), likely to be biased to producers of higher-quality cement.

The clinker ratio in China has been below 0.8 since at least 1990, and has declined rapidly in the last decade to about 0.62 in recent years (Figure D3). Along with the use of clinker substitutes mentioned above, the use of modern kiln types also contributes. The New Suspension Preheater (NSP) type, which allows lower clinker ratios to be used in cement production given the same strength requirements, was used for about one-seventh of production in 2000, a share which had grown to about four-fifths in 2010 (Xu et al., 2012).



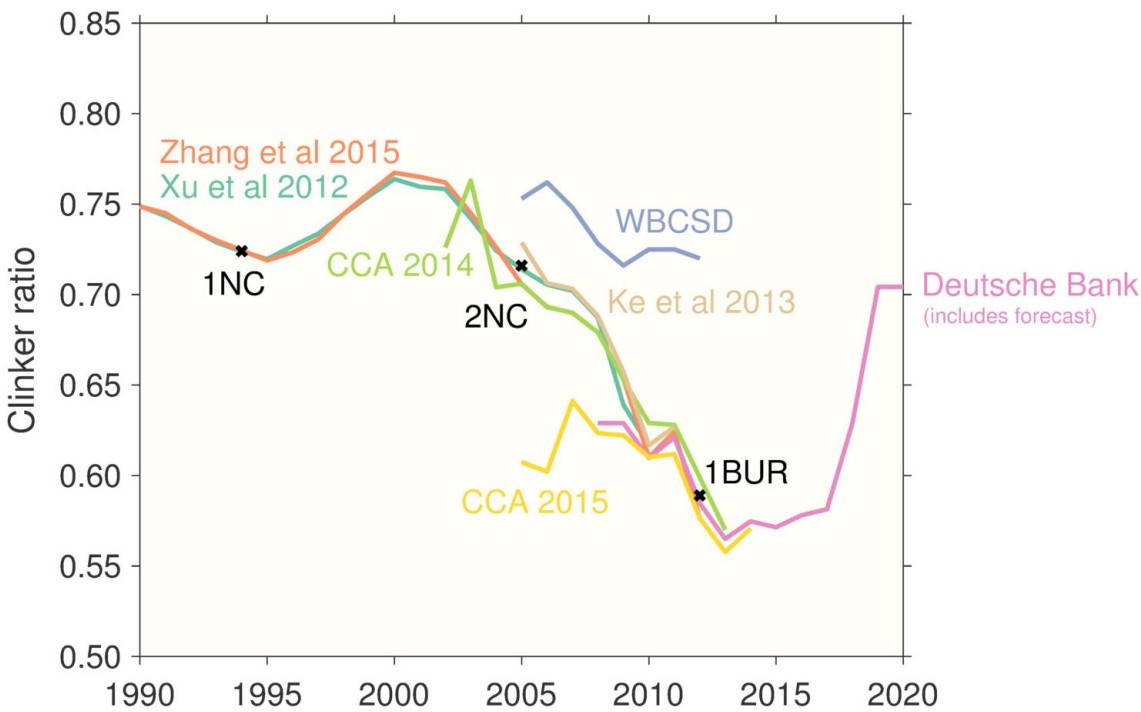

**Figure D3: China's clinker ratio since 1990, from a number of different sources. The three official estimates are marked in black: 1NC is the First National Communication, 2NC the Second National Communication, and 1BUR is the First Biennial Update Report.**

The default factor for the average lime (CaO) content of clinker given by the IPCC 2006 guidelines is 65%. Liu et al. (2015) used 62%, being the weighted average derived from the factory-level study made by Shen et al. (2014)[6]. However, clinker production also involves the decomposition of $MgCO_3$ to MgO, and emission factors derived only from the CaO content (including Liu et al., 2015) omit this source of $CO_2$ emissions, which Annex-I Parties include in their inventories.

China's Second National Communication used emission factors 'derived from in-situ surveys' (p60), while the First Biennial Update Report using factors 'obtained through typical enterprise survey' (p19). The factor used for the Second National Communication is provided in the NDRC's report: 0.5383 (NDRC, 2014). This factor excludes clinker kiln dust, stated to be negligible, but does include emissions from the decomposition of $MgCO_3$.

For years before 1990, the assumption is made here that the clinker ratio was 0.8 until 1970, and then linearly declined to the estimated value in 1990.

The cement emissions derived in this study are shown in Figure 3, which also compares with several other available estimates. The 2011 dip in cement emissions presented by Liu et al. (2015) appears to be spurious, based on an unlikely low clinker ratio of 0.49 in that year. Recent data from CCA indicate a ratio of 0.63 in that year, with no particular discontinuity.

---

[6] Confirmed by Z. Liu, personal communication, 2017.



### D3 India

India is the second-largest producer of cement in the world, with about 270,000 tonnes in 2017 (USGS, 2018). The 47% of India's cement production covered by WBCSD's data used a clinker ratio of 0.70 in 2014 (WBCSD, 2012).

In India's First National Communication to the UNFCCC, with data for 1994, process emissions from cement production are reported as 30767 $ktCO_2$, using an emission factor of 0.537 $tCO_2$/t clinker (p41), implying clinker production of 57294 kt in that year (Ministry of Environment & Forests, 2004). USGS reports Indian cement production in that year as 57000 kt. Allowing for rounding, the implied clinker ratio was therefore surprisingly high at approximately 1.0 in 1994. WBCSD data indicate that the clinker ratio in 1990 was 87% for the cement manufacturers from which they had data (WBCSD, 2014). These data are inconsistent, but it is unclear where the error lies.

Similarly, in India's Second National Communication, with data for 2000, process emissions are reported as 44056 $ktCO_2$, using the same emissions factor (p53), implying clinker production of 82041 kt (Ministry of Environment & Forests, 2012). USGS reports cement production in 2000 of 95000 kt. The clinker ratio was therefore most likely about 0.86 in 2000, agreeing closely with that reported by WBCSD (0.85).

India's first Biennial Update Report reports cement process emissions of 83851.74 ktCO2 in 2010 (Ministry of Environment Forest and Climate Change, 2015). Energy emissions were reported to have been about the same as in 2000 implying vastly improved efficiency. The BUR does not indicate what emission factor they've used, but assuming 0.537 as before would suggest 156 Mt clinker production in 2010.

With no complete official time-series of either clinker production or clinker ratio, a multi-source approach has been used here. We make use of data from the Indian Cement Manufacturers' Association (CMA), consultancy reports from CRISIL and IBEF, WBCSD, and other sources. Data include clinker production, blending ratio (the inverse of clinker ratio), and cement types. When calculating clinker ratios from clinker and cement production data, clinker trade has been taken into account.

The cement-type data (OPC, PPC, etc.) indicates a dramatic shift to OPC, between 1986 and 1990, suggesting an improvement in quality. This appears to have been a result of 'decontrol' in 1989, which removed many regulations from the industry. Since 2000 the cement types have begun to change again, a result of growing acceptance of other types of cement as being of sufficient quality (CRISIL, 2017, p. 20).





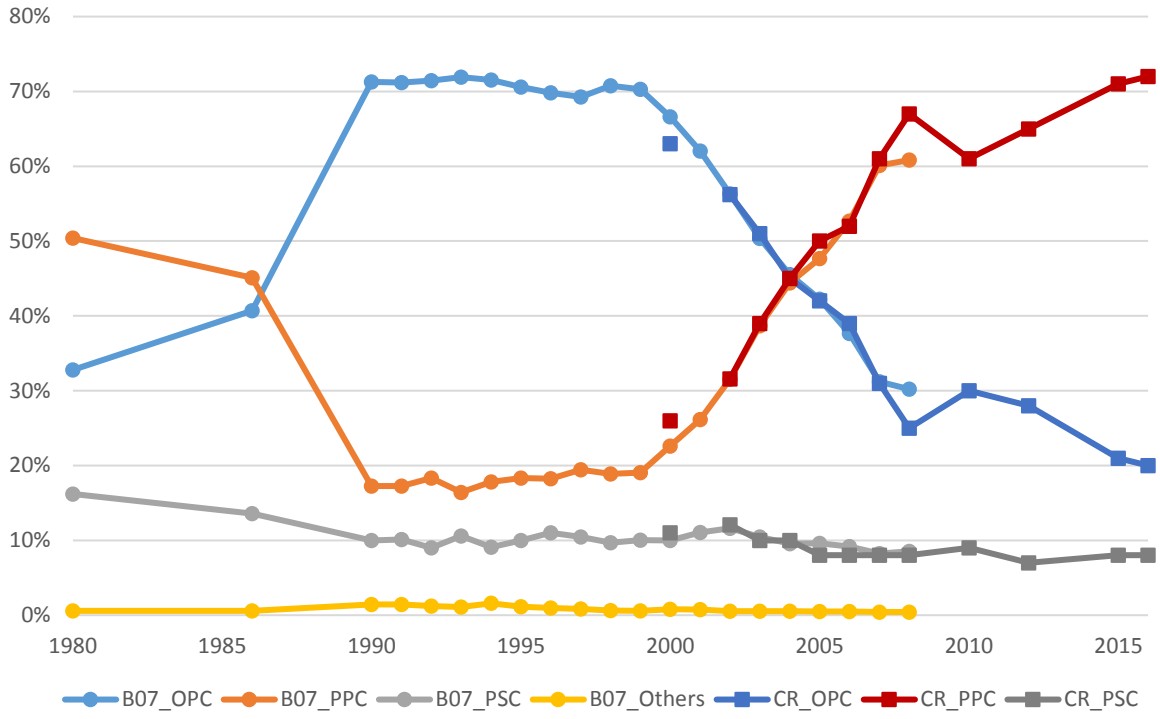

**Figure D4: Proportions of cement production by type. B07: (J.D.Bapata et al., 2007), CR: (CRISIL, various years). OPC: Ordinary Portland Cement, PPC: Portland Pozzolana cement, PSC: Portland Slag Cement.**

Using the cement types, combined with the proportion of clinker in each cement type, one can derive the overall clinker ratio

from a weighted average. The proportions of clinker in each cement type change over time, and only two sets of estimates were available: one from the WBCSD and IEA (2013), assumed to represent 2012 and later, and another from IBEF (2005), assumed to represent 2005 and earlier. The clinker ratios by cement type were interpolated linearly between these two years. The WBCSD survey data for India cover close to half of Indian cement manufacture. These data show that the clinker ratio has declined from 0.86 in 1990 to 0.70 in 2014.

Various reports on the Indian cement industry by consultancy CRISIL give data on both clinker production and blending ratio for various years.

The CMA also provides clinker production data, but in the 2009-10 financial year two members discontinued their membership of the Association, so production data from that year onwards are incomplete (CMA, 2010).

There unfortunately remains some disagreement between the clinker ratios derived from different sources (Figure D5). The

data from the WBCSD represent just under half of cement production in India, most likely the larger producers. There is a significant divergence in 2009/10 between WBCSD and the other data sources. CRISIL reports that "the blending ratio dipped significantly to around 1.25 from 1.34 in 2008-09. Cement players had lowered the blending ratio during the year on account of decline in cement demand and increased clinker production." (CRISIL, 2013, pA-19). The cement-type data also show a





sharp increase against the trend in the amount of OPC produced at that time, from 25% in 2007-08 to 30% in 2009-10. It may be that the survey-based approach of WBCSD did not capture this adjustment in the industry.

The use of clinker production data is clearly preferred. When clinker production data were not available in earlier years, we have used the analysis based on cement types. In later years we use the reported blending ratios (reciprocal of the clinker ratio).

5  Data were adjusted from financial to calendar years by using monthly cement production data.

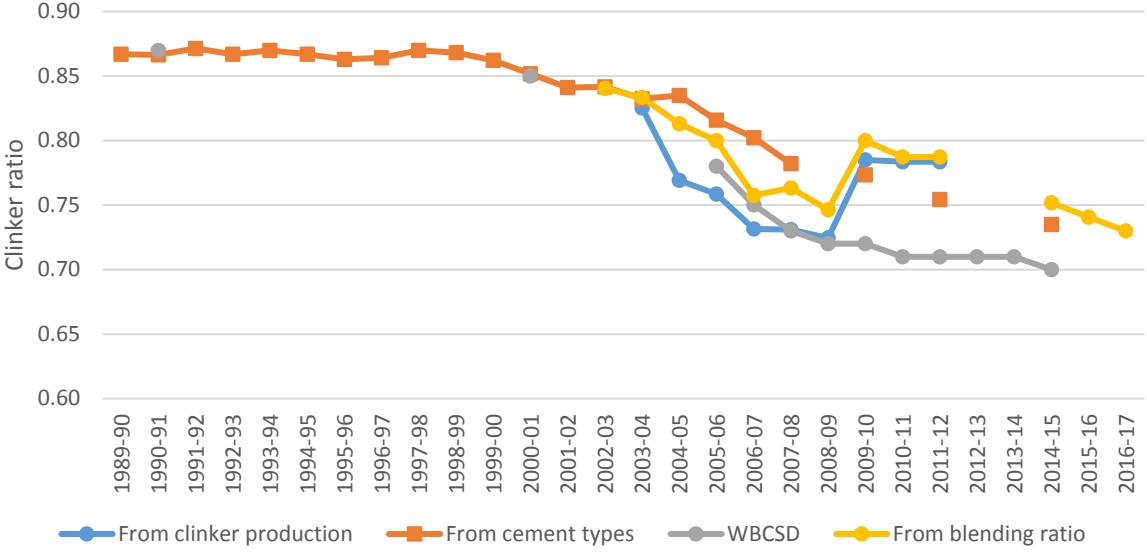

**Figure D5: Estimates of clinker ratio in India from various sources.**

The clinker ratio must be applied to cement production data, but there is some divergence between USGS data and those from the Office of the Economic Advisor (OEA), which are reported by the CMA (Figure D6). This divergence has not yet been

10  explained. In this work we rely on the official data from the OEA, although this only affects the emissions estimate after 2016, because clinker production estimates are used for 2004–2015.



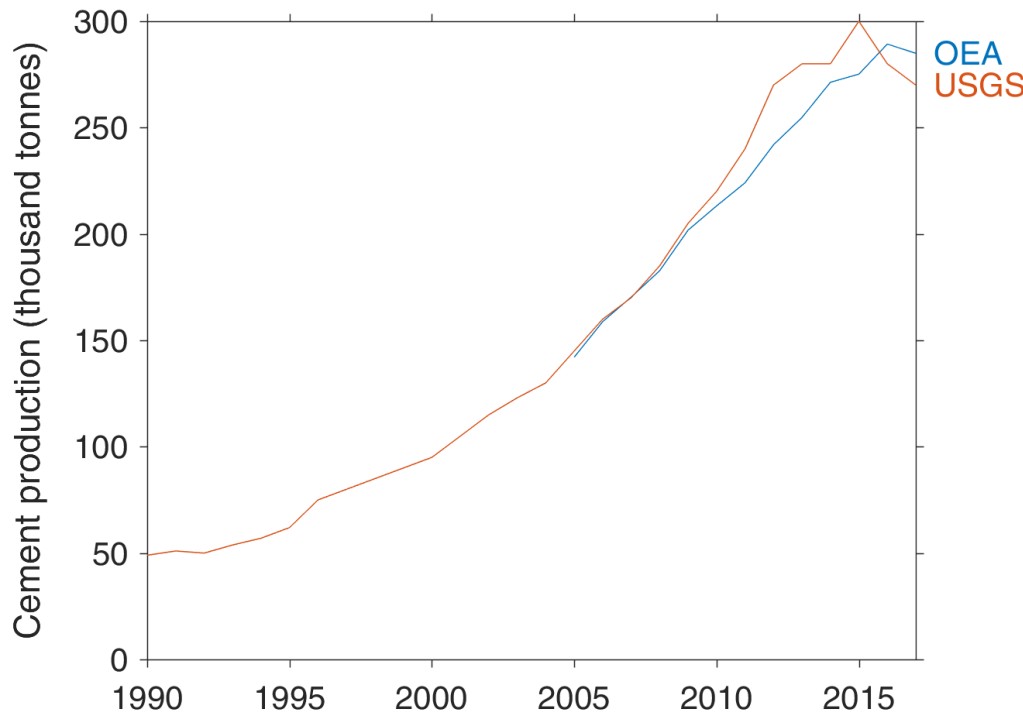

**Figure D6: Comparison of Indian cement production data from USGS and OEA, the latter beginning in 2005.**

Indian analyses have shown emission factors (t $CO_2$ (t clinker)$^{-1}$) similar to the default IPCC factor of 0.52 (Arceivala, 2014), so we use that factor here.

5  The final emissions time-series lies very close to the three available official estimates (Figure D7).



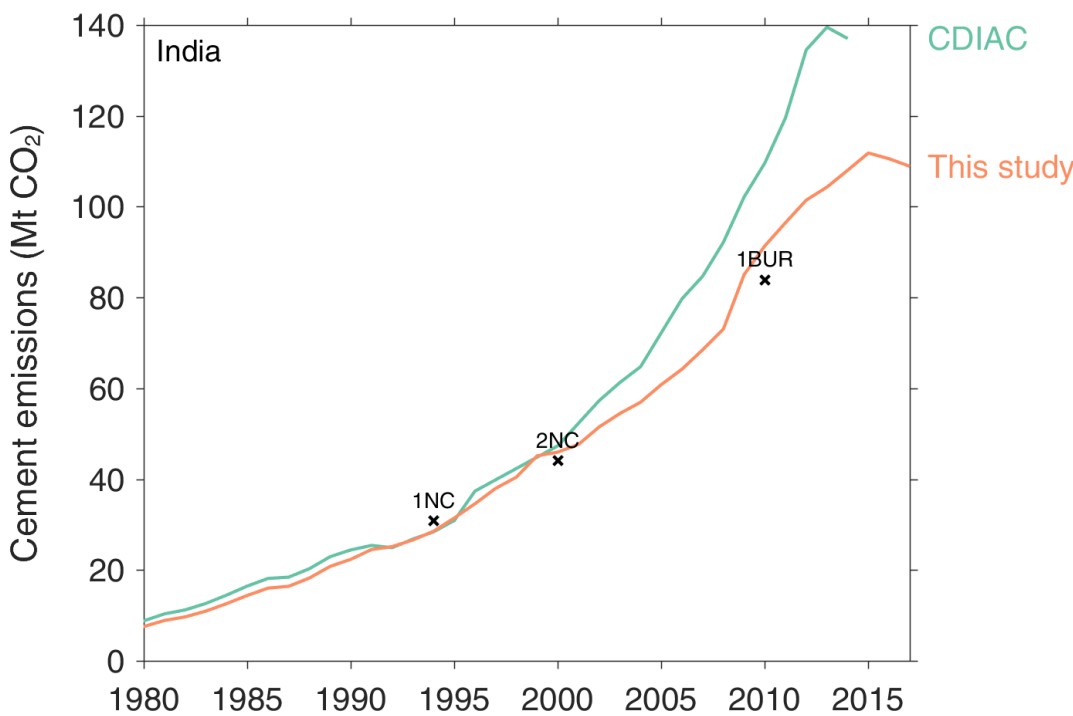

**Figure D7: Revised cement emissions for India. 1NC: First National Communication; 2NC: Second National Communication; 1BUR: First Biennial Update Report.**



## D4 USA

The USA reports annual emissions from cement production to the UNFCCC, along with all other Annex-I Parties. However, in addition to this series, which starts in 1990, the US Geological Survey (USGS) have an unpublished time series of clinker production in the US starting in 1925 (Hendrik van Oss, USGS, personal communication, 2015). These allow very good
5    estimates of $CO_2$ emissions from historical clinker production. Furthermore, while USGS clinker data begin in 1925, the clinker ratio was very close to 1 between 1925 and 1970. By assuming that it was also 1 between 1900 and 1924, the data series can be extended back to 1900, when cement production data begin (Figure D8).

Until about 1970, CDIAC's estimates of US cement emissions show good correspondence with estimates calculated directly from clinker production data. However, after about 1970 significant deviations appear as the clinker ratio of US cement began
10   to drop below unity (Figure D8). The same method is used here to calculate emissions from clinker production data as is used in the US National Inventory Report. The reason for the divergence seen in Figure D8 is that the UNFCCC submission includes cement production in Puerto Rico, while the estimates in this study do not.

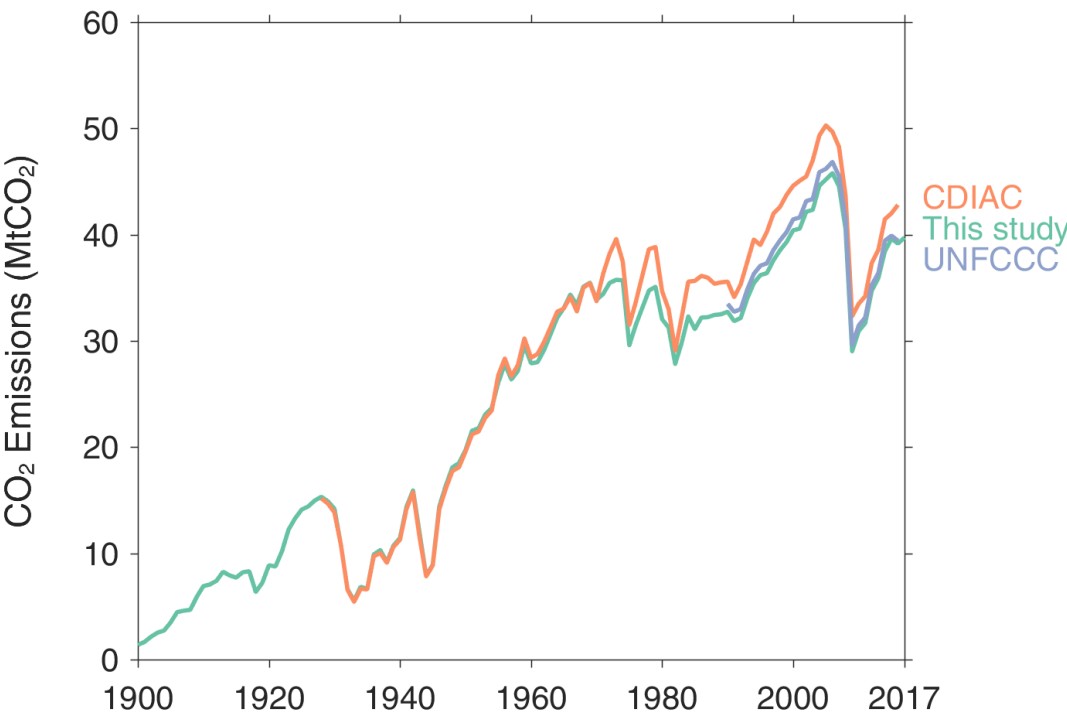

**Figure D8: Revised cement emissions for USA.**



## D5 Armenia

Armenia's 2010 National Inventory Report provides emissions from cement production for 1990–2010 (Ministry of Nature Protection, 2014). The implied emission factor is nearly constant, at around 0.507 every year. The second National Inventory Report for 2012 provides emissions for 2000-2012, now using Tier III methodology (Ministry of Nature Protection, 2015). These have been combined with the earlier estimates to give a longer data series from 1990-2012. The introduction of Tier III methodology raised emissions in the overlapping period by an average of 14%, and this was used to adjust the emissions from the first NIR.

Armenia's clinker production was significantly higher than USGS-reported cement production in 1990 and 1991, indicating significant exports or stockpiling of clinker in those years Figure D10. While clinker production dropped significantly below cement production in the following few years, there have been a number of years since when clinker appears to have been exported.

While it is quite possible that Armenia was a net exporter of clinker in years prior to 1990, no data have been found to substantiate this. After 2012 we assume that the ratio of clinker production and cement production in 2012 continue, with the emission factor of 2012.

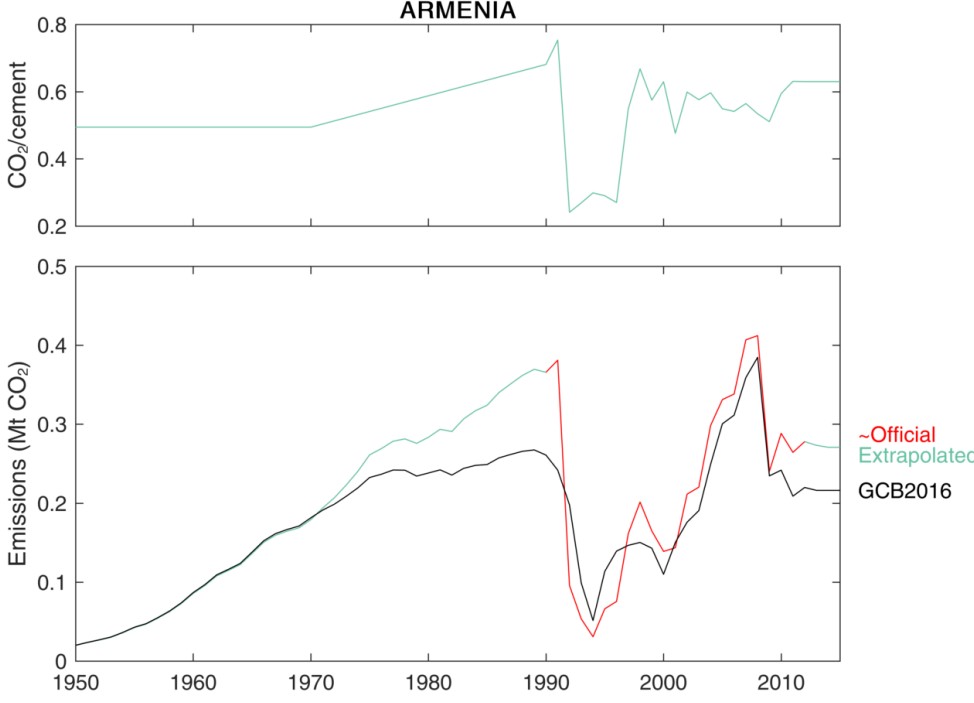

**Figure D9: Revised cement emissions for Armenia.**




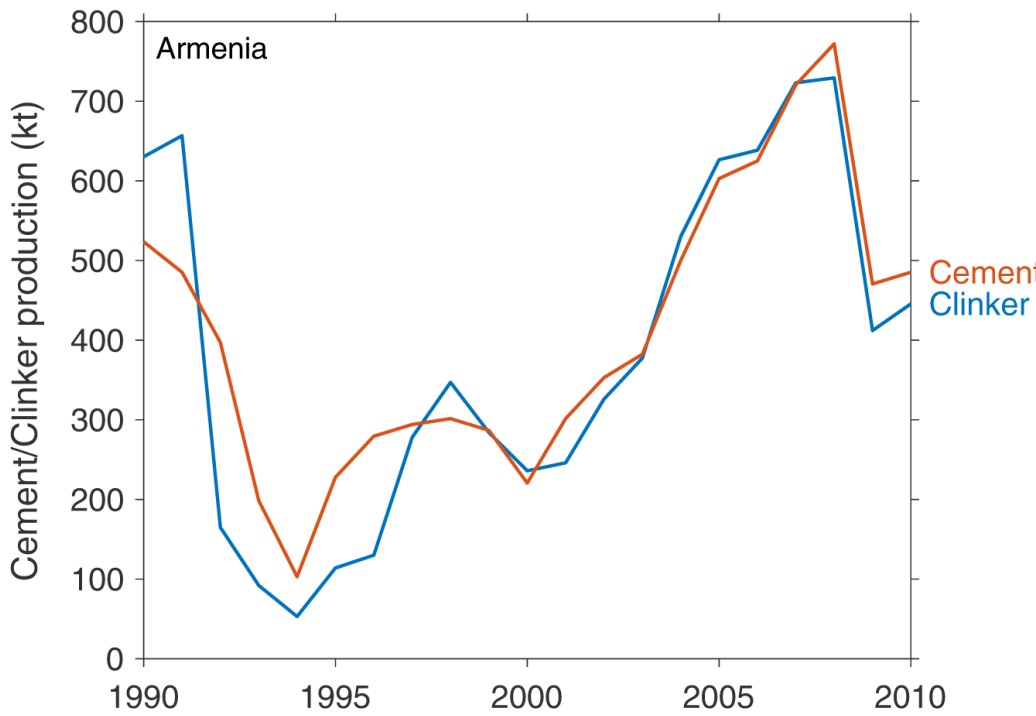

**Figure D10: Clinker and cement production in Armenia, 1990–2010 (Ministry of Nature Protection, 2014; van Oss, 1994–2018).**




## D6 Azerbaijan

Azerbaijan's Third National Communication provides estimates of emissions from cement production for 1990, 2000, and 2005–2012.

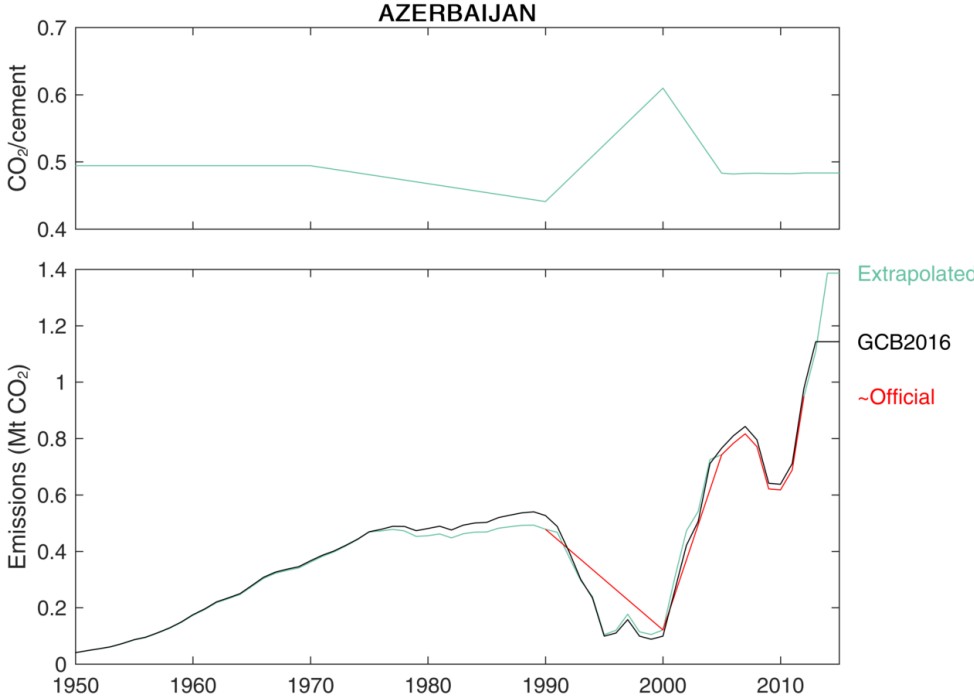

5    **Figure D11: Revised cement emissions for Azerbaijan.**




## D7 Bosnia and Herzegovina

Bosnia-Herzegovina's Second Biennial Update Report includes a chart showing estimates of cement emissions for 2002–2013, and specific estimates are provided in the text for 2003 and 2012.

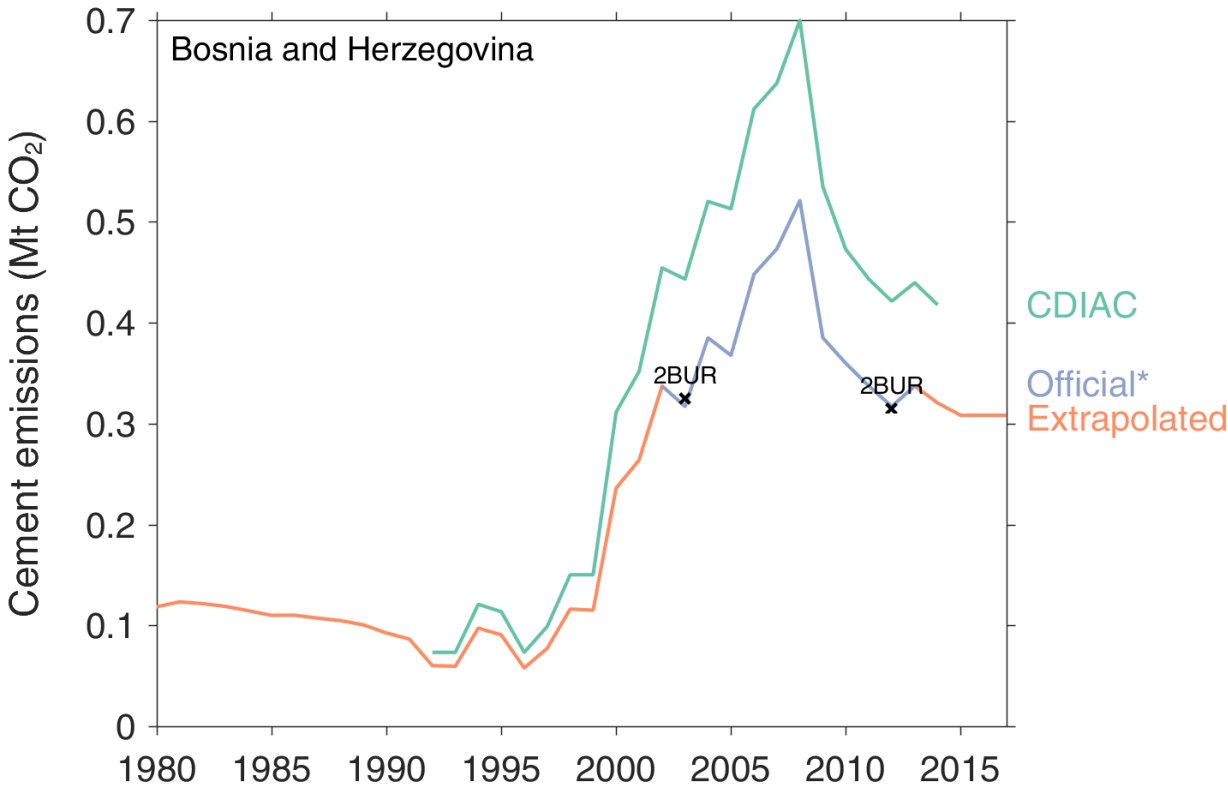

**Figure D12: Revised cement emissions in Bosnia and Herzegovina.**

## D8 Brazil

Brazil's Third National Communication to the UNFCCC includes estimates of emissions from cement production from 1990 to 2010 (MSTI, 2016). The emission factor ranges between 0.544 and 0.549 t $CO_2$ (t clinker)$^{-1}$, for the years where clinker production data are provided. The clinker ratio (assuming zero clinker trade) has declined from 0.78 in 1990 to 0.66 in 2010 (Figure D14).

The report states that Brazil has been substituting clinker in cement manufacture "for over fifty years" (p100). For years before 1990, clinker ratio was interpolated linearly between 0.95 in 1965 to the estimated ratio in 1990 from the data. After 2010, the clinker ratio was assumed constant at the 2010 level.




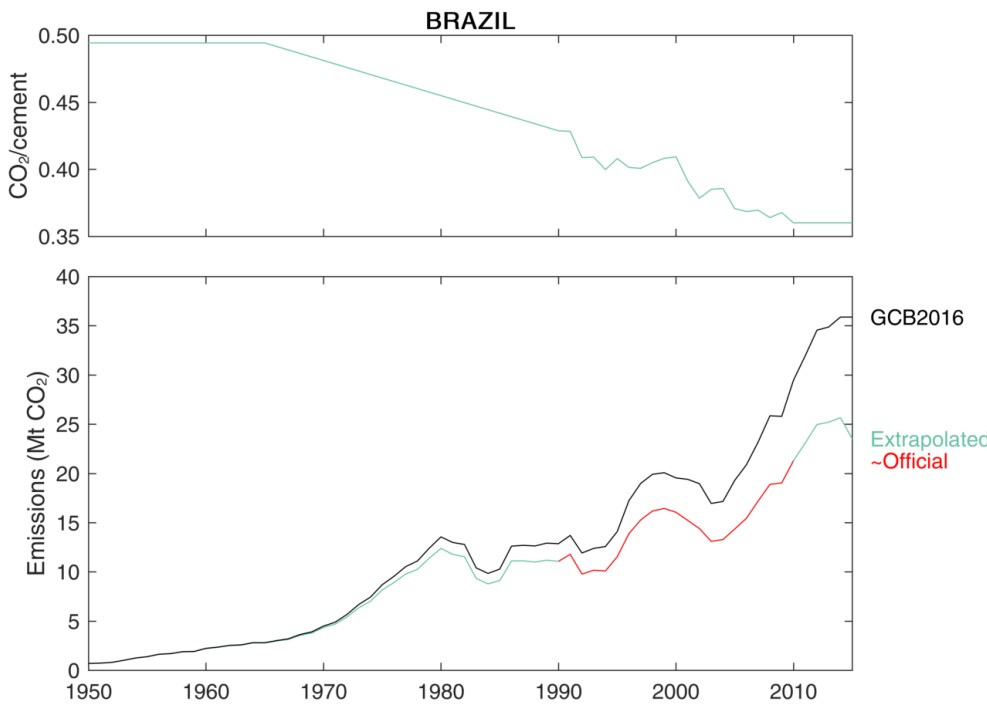

**Figure D13: Revised cement emissions for Brazil.**



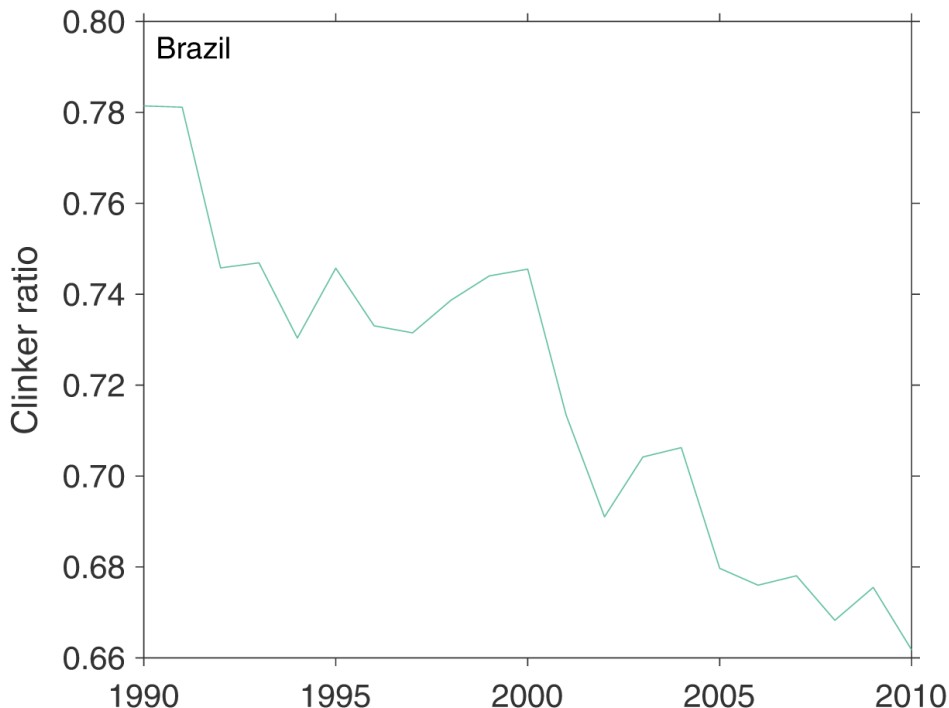

**Figure D14: Brazil's approximate clinker ratio, with no account for clinker trade.**



## D9 Chile

The Chilean National Inventory Report (MdMA, 2017) presents clinker production data for 1990–2013, with 1990–1994 and 2013 estimated based on extrapolated clinker ratios. The country uses IPCC default emission factors in the absence of country-specific data. Significant imports of clinker mean that the resulting emissions are significantly lower than those estimated by

5    CDIAC (Figure D15).

Imports were negligible in 1990, so an assumption has been made of no imports prior to 1990. For years after 2013, the ratio of clinker production to cement production has been assumed to continue, implicitly assuming the same clinker ratio and clinker trade ratios.

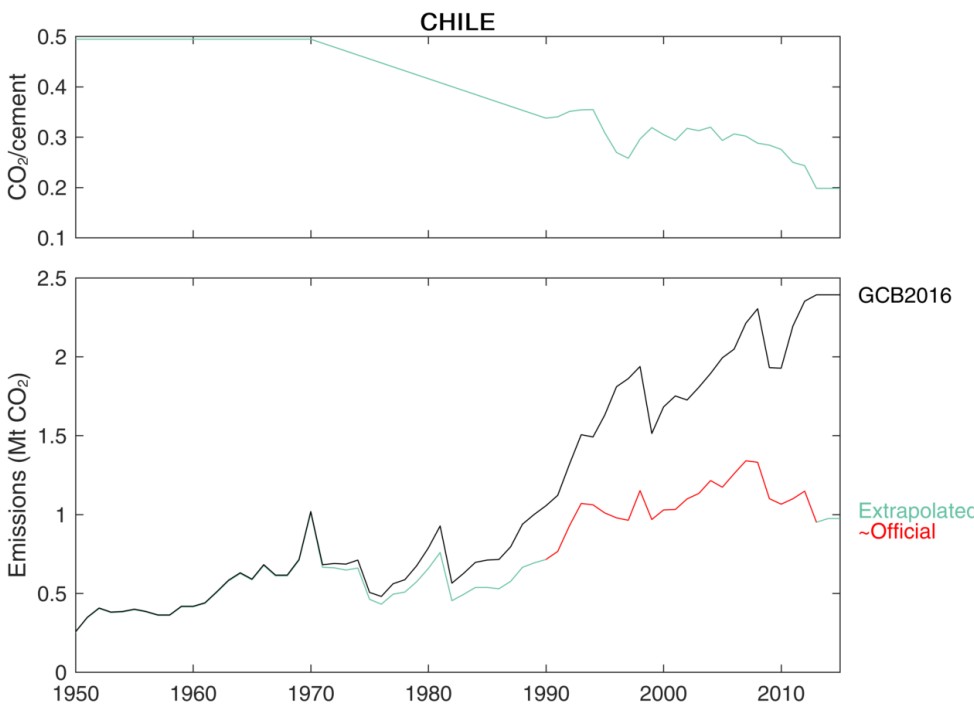

10    **Figure D15: Revised cement emissions for Chile.**



## D10 Colombia

Colombia's First Biennial Update Report includes a chart showing estimates of cement emissions for 1990, 1994, 2000, 2004, 2010 and 2012.

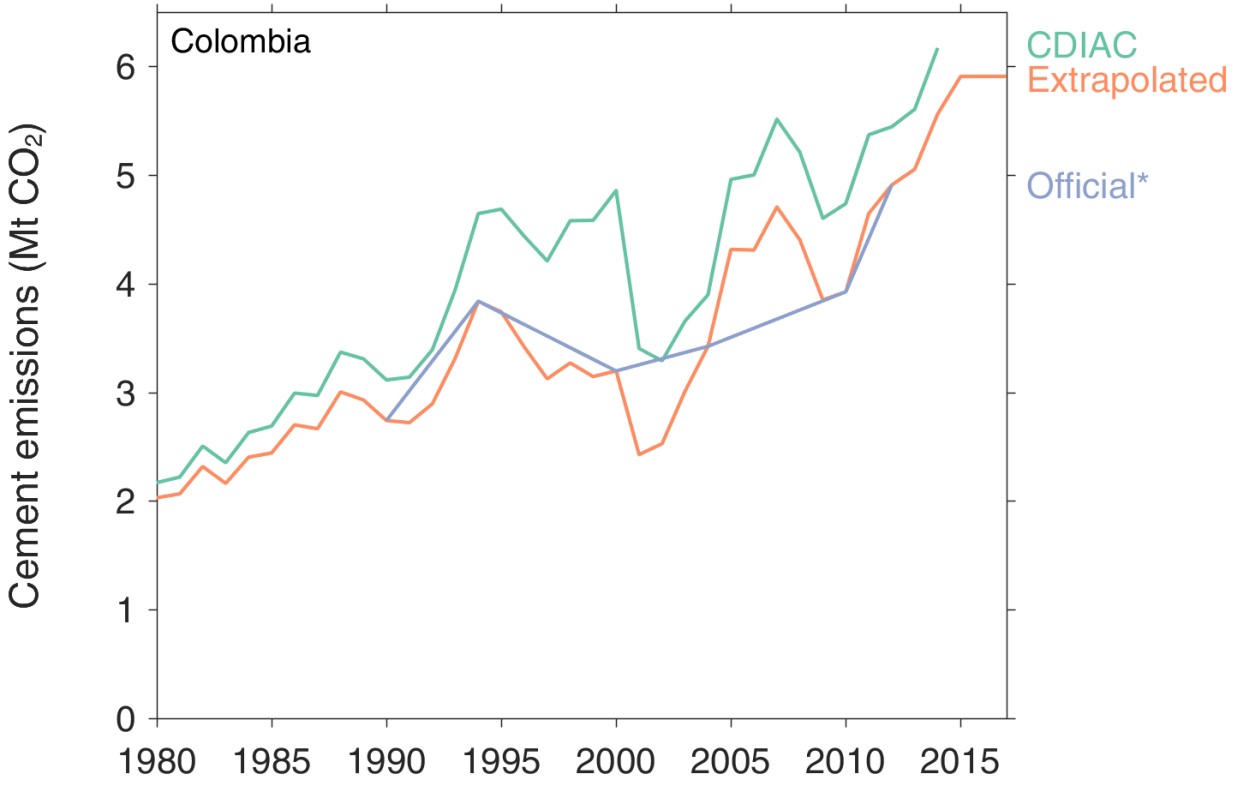

5    **Figure D16: Revised cement emissions in Colombia.**

### D11 Indonesia

Indonesia's First Biennial Update Report provides estimates of process emissions from cement production for 2000–2012, using the IPCC default emission factor. Clinker production is higher than cement production in many years.





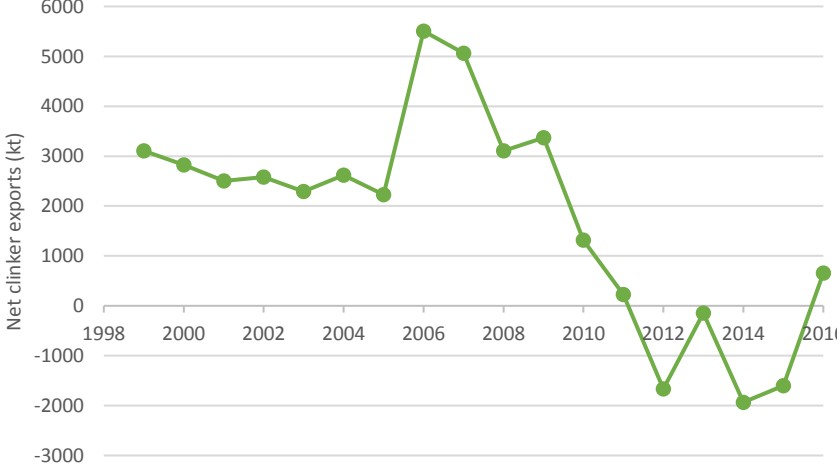

**Figure D17: Revised cement emissions for Indonesia.**

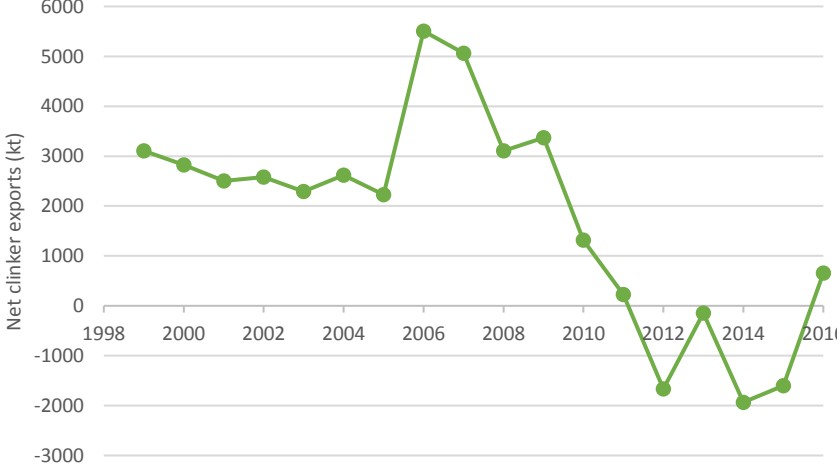

**Figure D18: Net clinker exports from Indonesia, 1999-2016 (Source: Statistics Indonesia).**





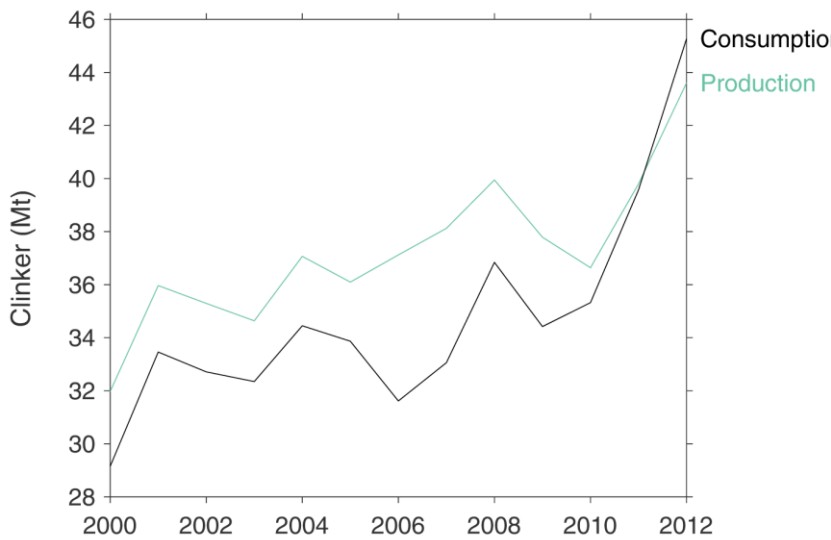

**Figure D19: Indonesian clinker production and derived consumption, 2000-2012.**

The clinker ratio, even after adjustment for clinker trade, is still above unity in some years, which is impossible (Figure D20).
This uses cement production data from USGS. Clearly there are some inconsistencies in the datasets used, and without clinker
production data it appears impossible to extrapolate a reasonable time series of cement emissions for Indonesia.

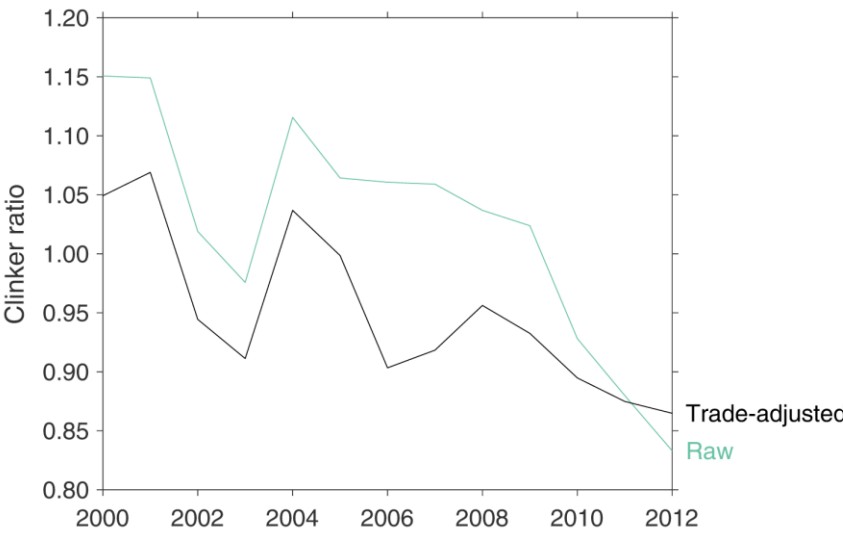

**Figure D20: Indonesian clinker ratio, calculated from both clinker production and consumption data.**



## D12 Israel

Israel's Third National Communication provides estimates of emissions from cement production for 1996, 2000, and 2003–2015.

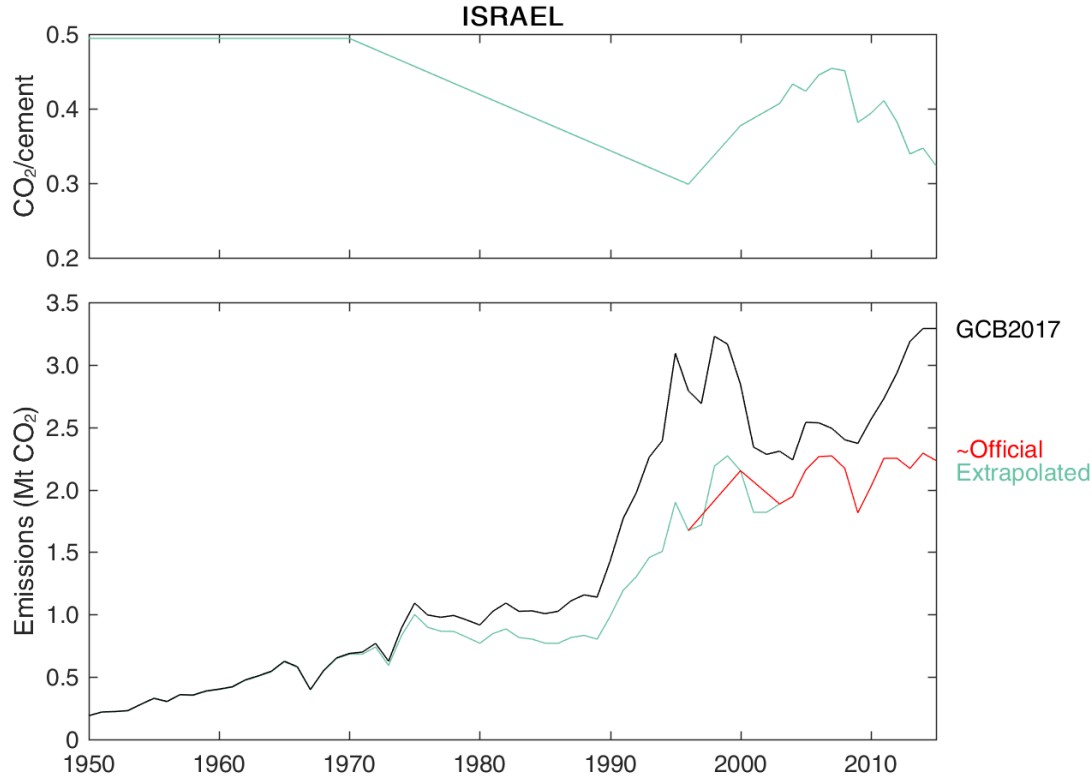

**Figure D21: Revised cement emissions in Israel.**

## D13 Jamaica

Jamaica's First Biennial Update Report presents clinker production and emissions estimates for 2006–12 (Mahlung and Dore, 2016). The implied emission factor used is 0.520 kg $CO_2$ (kg clinker)$^{-1}$.

The BUR states that clinker production data were obtained from the Caribbean Cement Company. Accordingly, further clinker production data have been sourced from annual reports of the Caribbean Cement Company (Caribbean Cement Company, various years) to extend this series to 1995–2016 (Figure D22).

The clinker ratio was 0.96 in 1995. For years before 1995, a clinker ratio of 0.95 has been assumed with the same emission factor of 0.520.



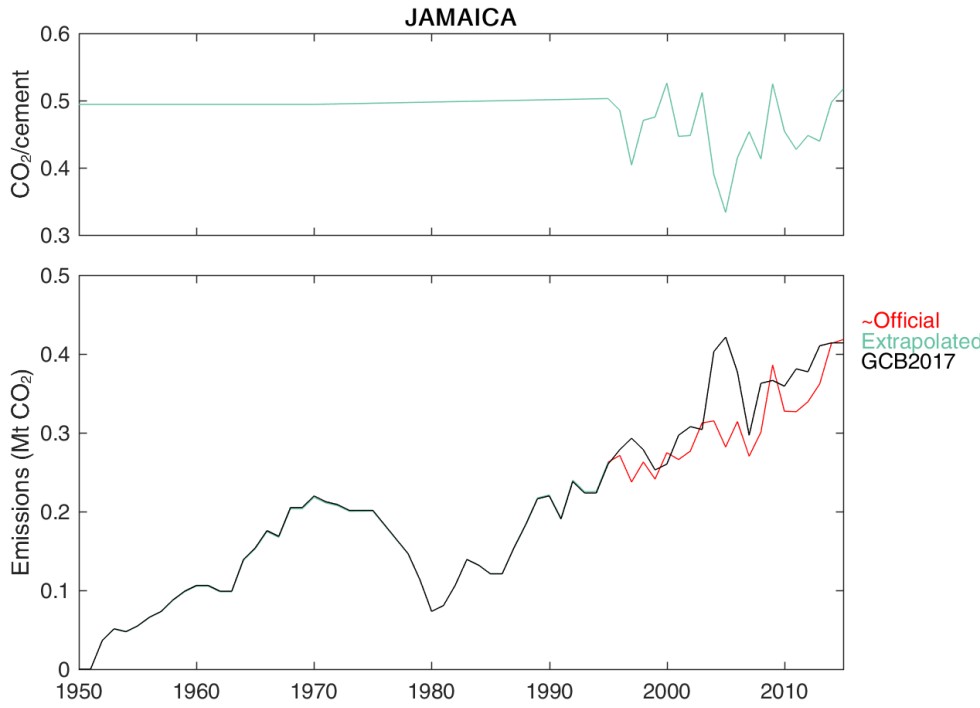

**Figure D22: Revised cement emissions for Jamaica.**





## D14 Korea

The Korea Cement Association (KCA) publishes annual national clinker and cement production from 1991, and at time of writing data were available to 2016.

The Third National Communication (Korean Ministry of Environment, 2012) states that cement production was 40.9% of total industrial process emissions of 56.7 Mt $CO_2$ in 2009, which comes to 23.19 Mt $CO_2$. Using an emission factor of 0.52 and the KCA clinker production figure of 44.774 Mt gives a very close 23.28 Mt $CO_2$ (Figure D23).

The clinker ratio over 1991–2015 from the KCA data show no clear trend, varying from year to year probably only in response to clinker trade (Figure D24).

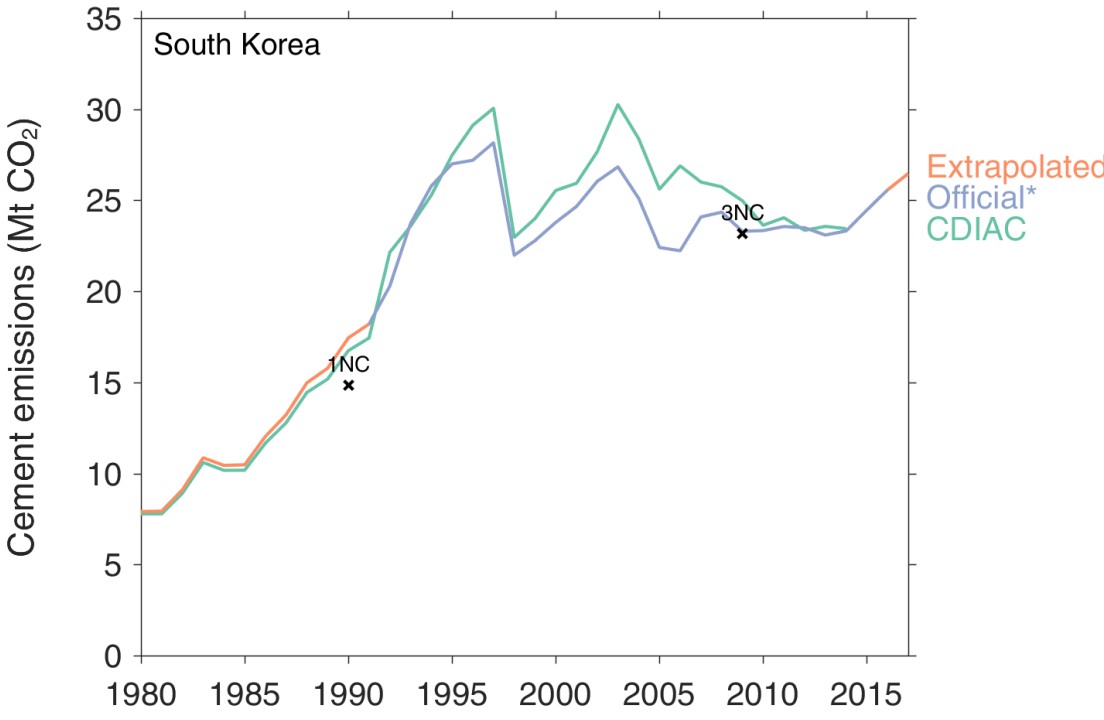

**Figure D23: Revised cement emissions for South Korea.**



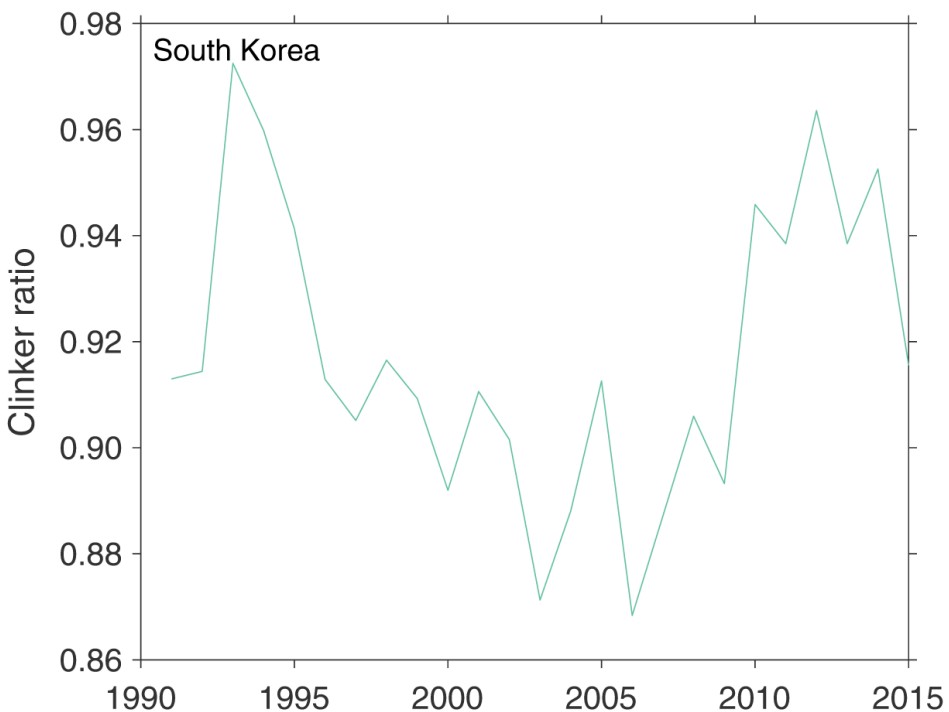

**Figure D24: South Korea's approximate clinker ratio, with no account for clinker trade.**



## D15 Lebanon

Lebanon's Second Biennial Update Report provides estimates of cement emissions for the years 1994, 2000, 2006, 2011 and 2013.

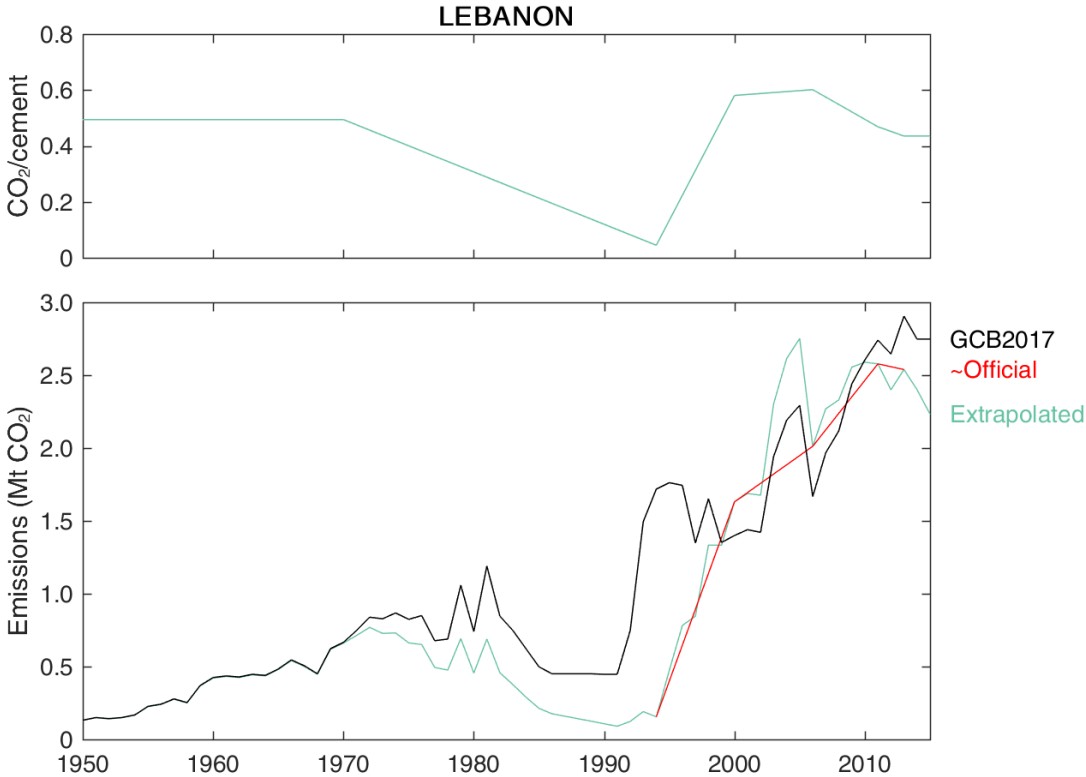

**Figure D25: Revised cement emissions in Lebanon.**

## D16 Mexico

Mexico's first Biennial Update Report (INECC and Semarnat, 2015) provides $CO_2$ emissions from cement manufacture 1990–2012 (Figure D26). Mexico has had significant clinker exports over this period, such that emissions are in many years higher than the estimates made by CDIAC.

After 2012, the emissions rate was assumed constant at the 2012 level, implicitly assuming constant clinker ratio and international clinker trade.





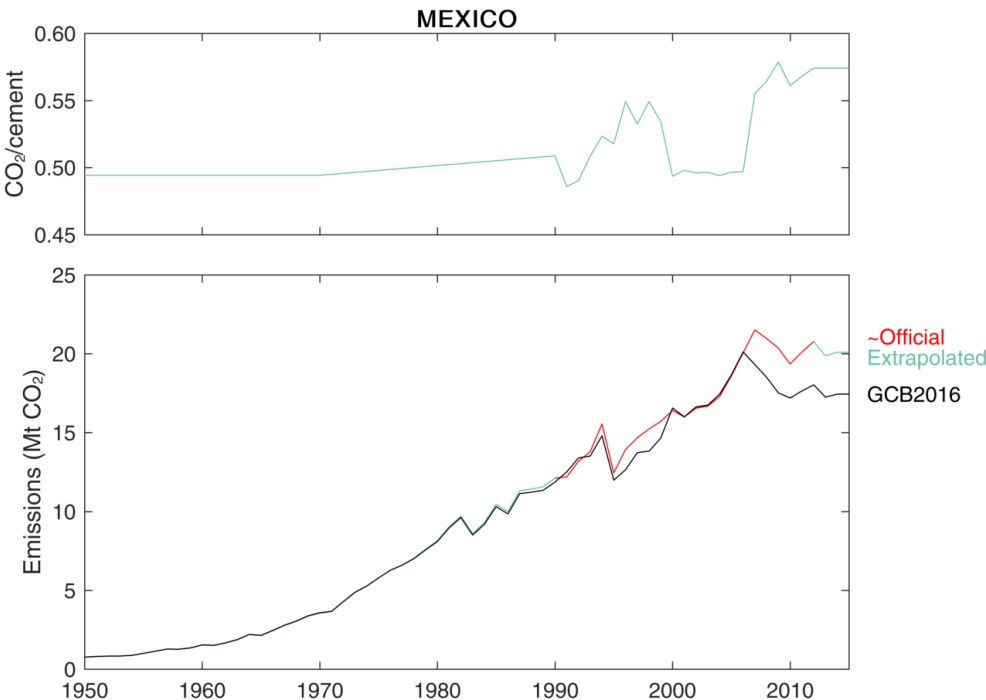

**Figure D26: Revised cement emissions for Mexico.**



## D17 Moldova

Moldova's National Inventory Report provides cement emissions for 1990–2012 (Ministry of Environment, 2013). Clinker production tracked cement production relatively closely over the entire period, although cement production was rather higher than clinker production in 1990, suggesting either exports of clinker or lower clinker ratio in that year (Figure D28).

5    After 2010 we assume that the ratio of clinker production and cement production in 2010 continue, with the emission factor of 2010 (Figure D27).

The main reason GCB2016 estimates were so low is that the method used to disaggregate emissions from countries of the Soviet Union assumed that the shares in 1992 represented the shares before 1992.

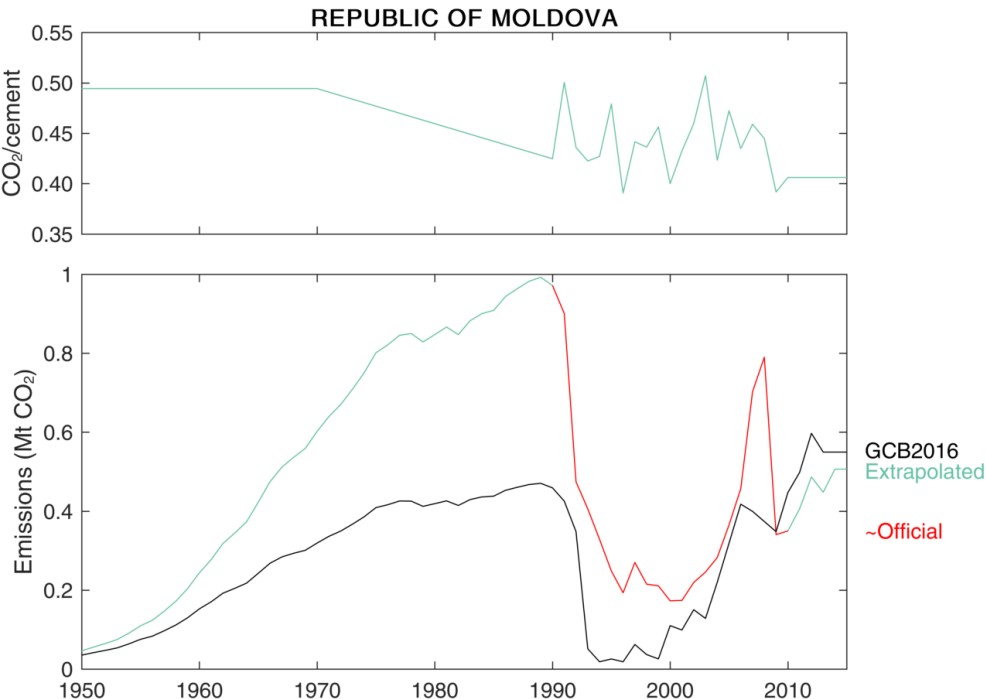

10    **Figure D27: Revised cement emissions for Moldova.**



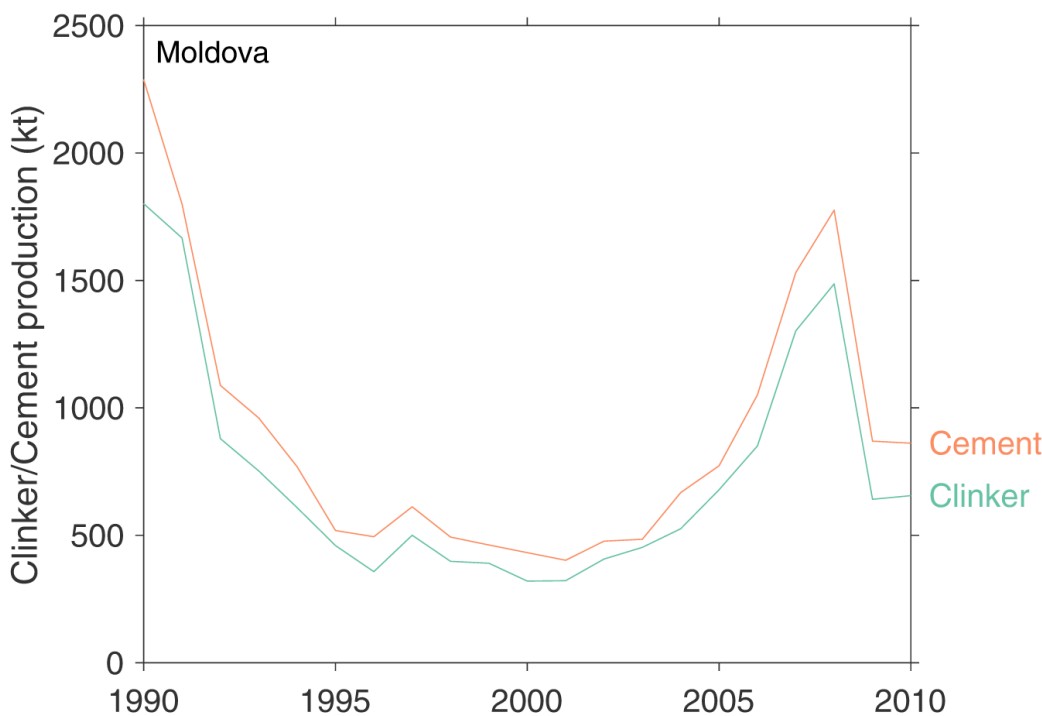

Figure D28: Clinker and cement production in Moldova (Ministry of Environment, 2013).



## D18 Mongolia

Mongolia's 2017 National Inventory Report provides cement emissions estimates for 1990–2014. The report also states that the first cement plant in Mongolia began operation in 1968.

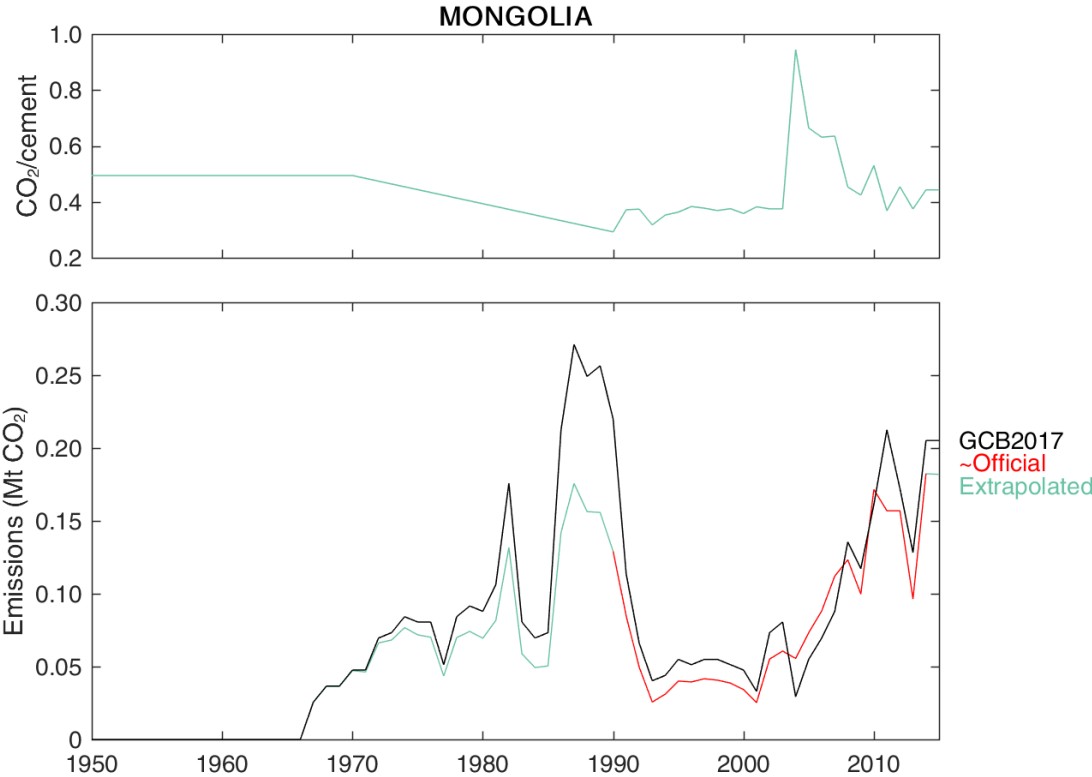

5    **Figure D29: Revised cement emissions for Mongolia.**

## D19 Morocco

Morocco's First Biennial Update Report provides estimates of cement emissions for 1994, 2000, 2005, 2006, 2008, 2010, and 2012.



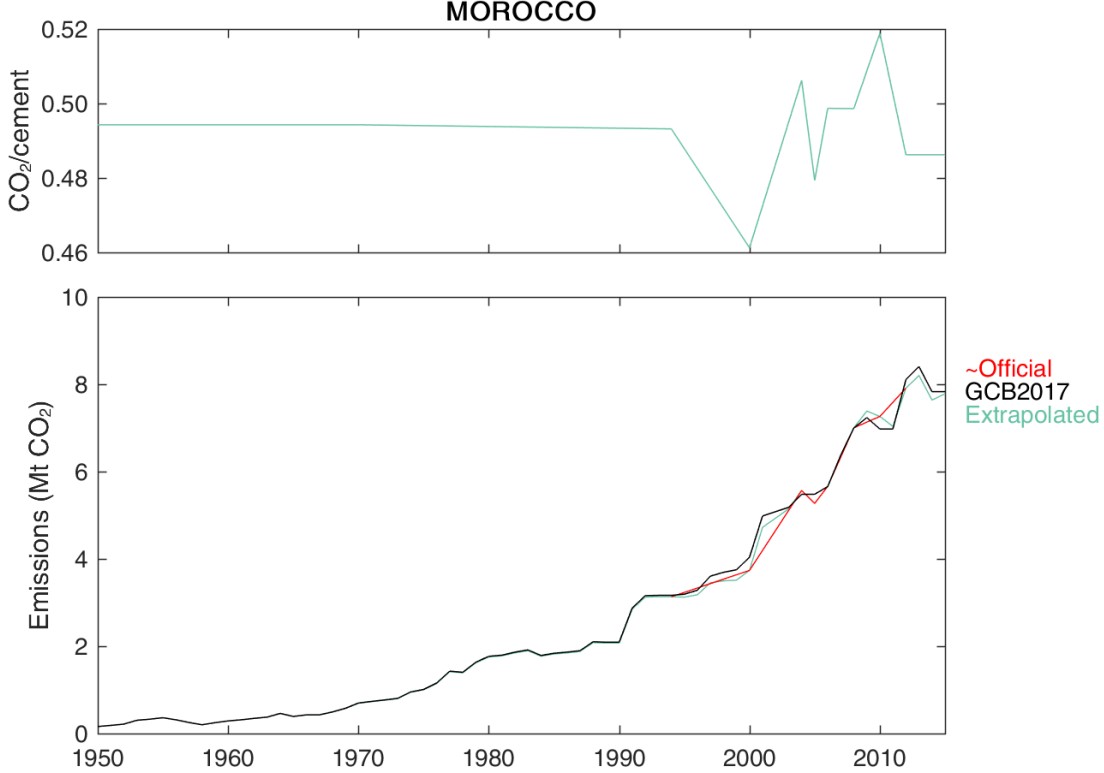

**Figure D30: Revised cement emissions for Morocco.**

### D20 Namibia

Namibia's second National Inventory Report provides estimates for emissions from cement production for 2000–2012, and clearly states that there was no cement production in the country before 2011.



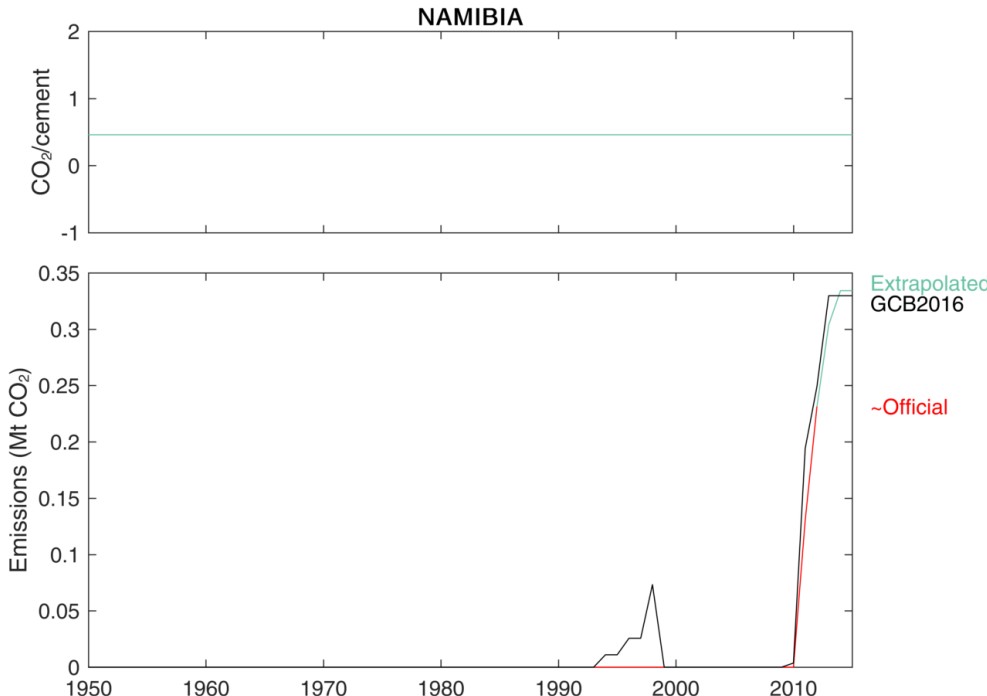

**Figure D31: Revised cement emissions for Namibia.**



**D21 Saudi Arabia**

The Saudi cement company Yamama Cement publishes national statistics of clinker and cement production (Yamama Cement, various years). Cement production statistics from USGS are largely consistent, with some exceptions, particularly USGS's most recent estimates for 2016 and 2017 (Figure D32). The Saudi cement market is currently characterised by significant

5    overproduction of clinker, with large stockpiles accumulating.

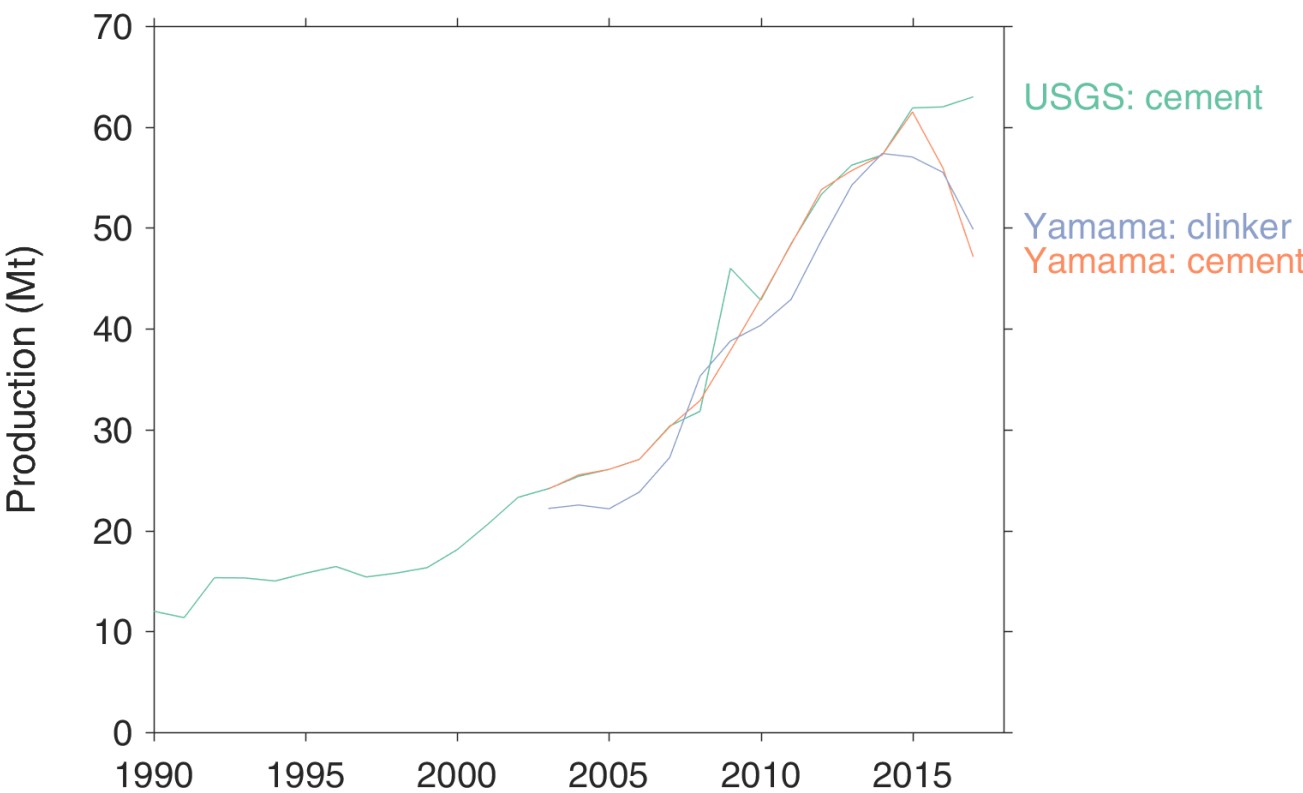

**Figure D32: Saudi Arabian cement and clinker production, 1990–2017 (Yamama Cement, various years; USGS, 2018; van Oss, 1994–2018).**

Saudi Arabia's three National Communications and first Biennial Update Report provide point estimates for cement emissions,

10    but when compared with clinker production data, the latter two suggest very high emission factors or disagreement in activity data (Figure D33). Neither the National Communications nor the Biennial Update Report provide any information on how emissions were calculated.



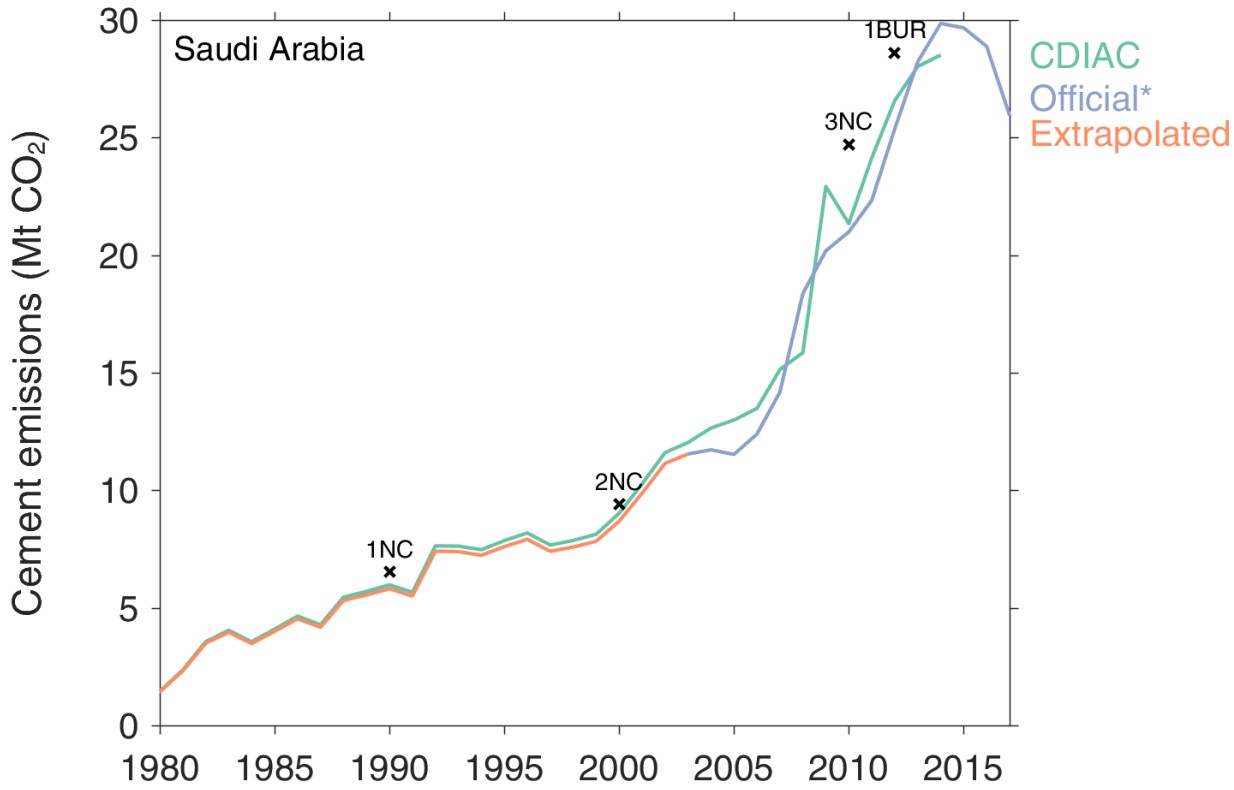

**Figure D33: Revised cement emissions for Saudi Arabia.**

**D22 Serbia**

Serbia's Second National Communication provides estimates of emissions from cement production for 1990, 2000, 2005 and
2010–2014.





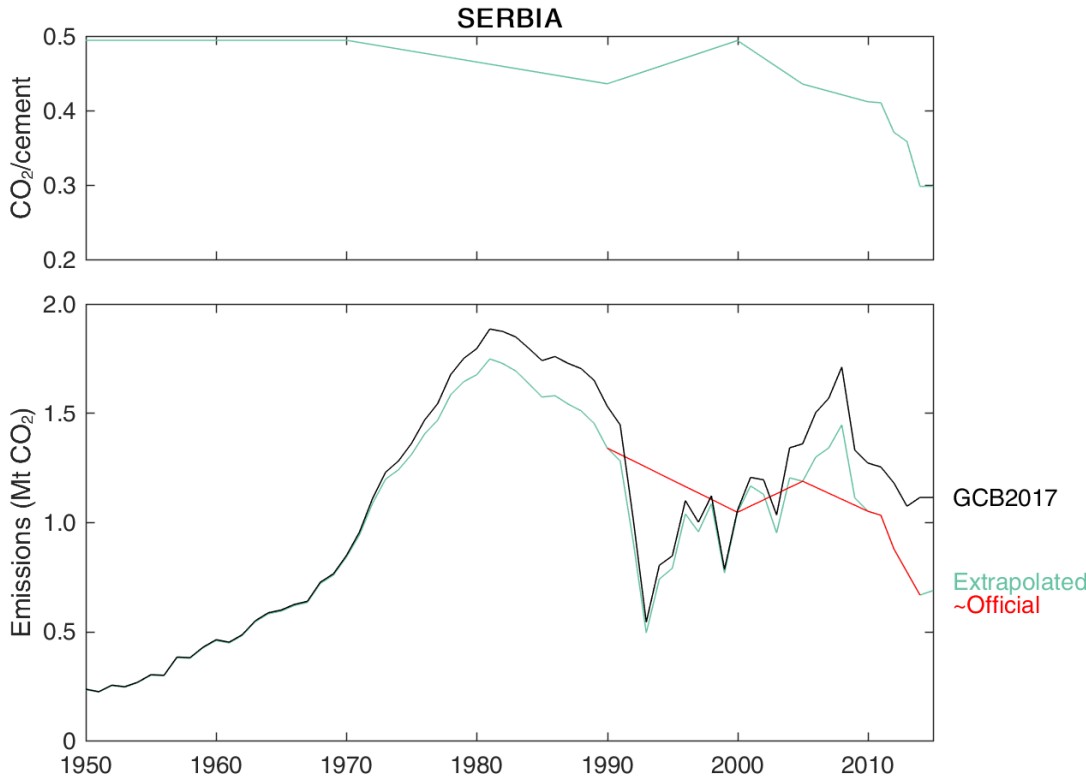

**Figure D34: Revised cement emissions in Serbia.**





## D23 South Africa

South Africa's first National Inventory Report (2014) provides estimates of emissions from cement production for 2000–2010.

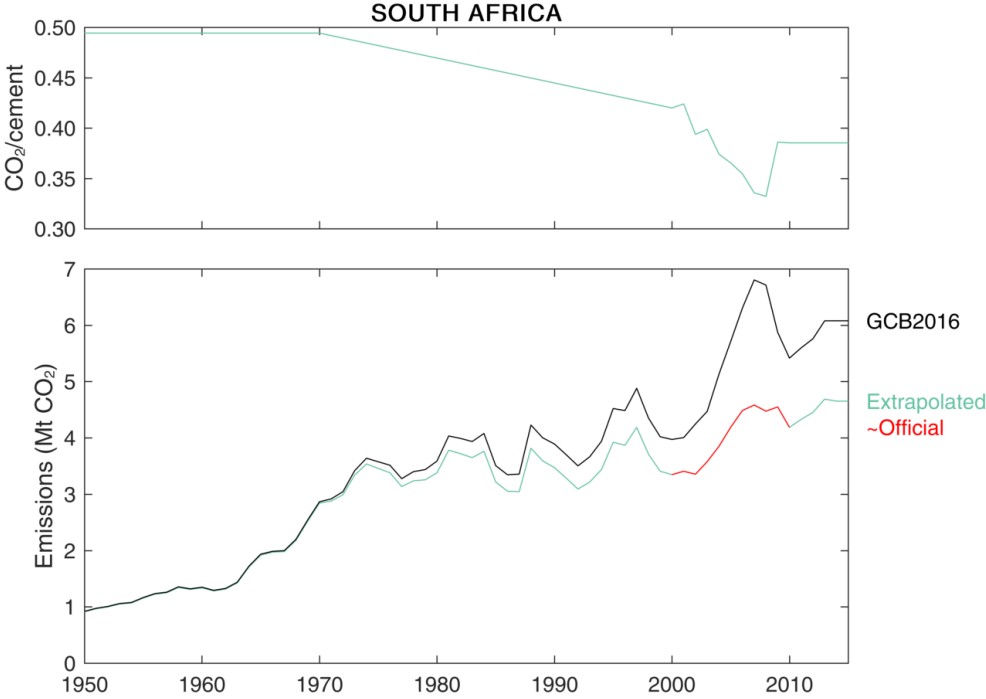

5 **Figure D35: Revised cement emissions for South Africa.**

## D23 Togo

Togo's First Biennial Update Report provides clinker production data and estimates of emissions from cement production for 1995–2015 (Ministry for the Environment and Forest Resources, 2017).





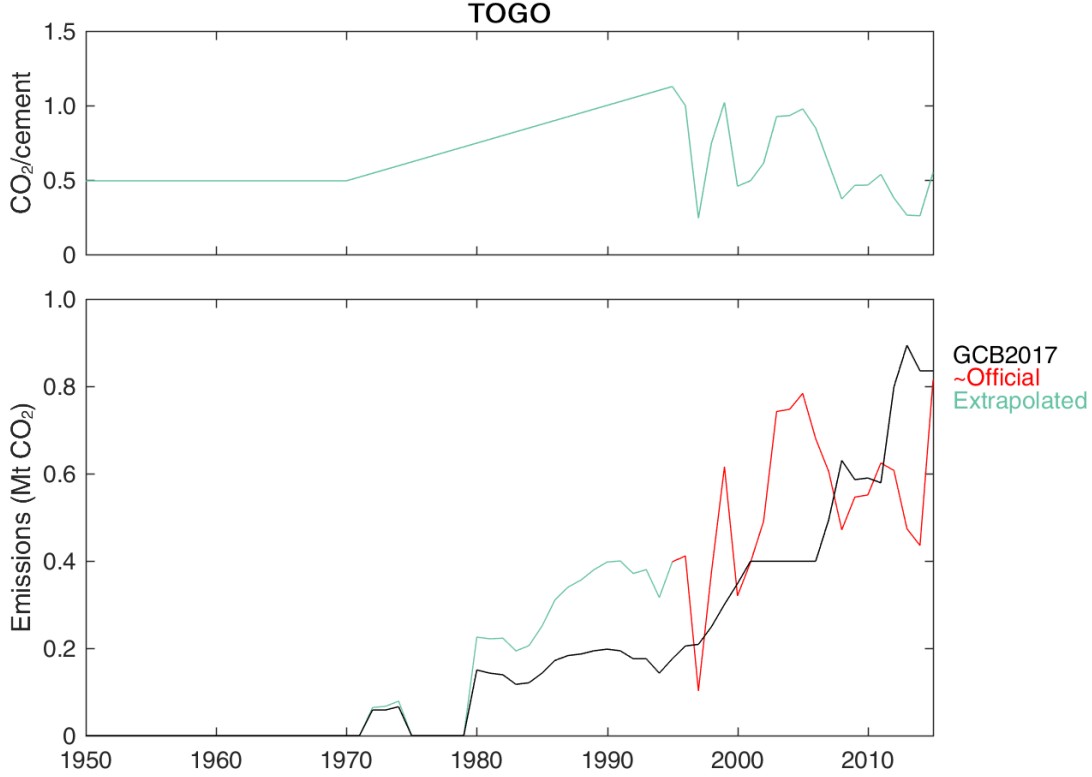

**Figure D36: Revised cement emissions in Togo.**

**D24 Uzbekistan**

Uzbekistan's National Inventory Report includes a time series of cement emissions for 1990–2012 (Uzhydromet, 2016).

5    After 2012, the emission factor and clinker ratio of 2012 were assumed constant (Figure D3).




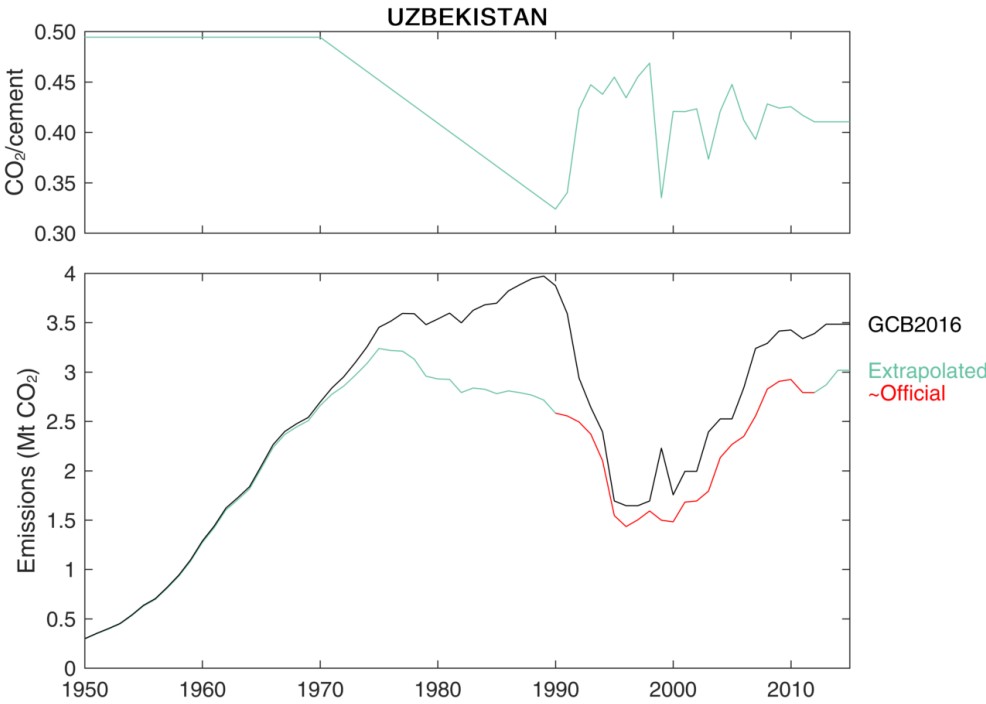

**Figure D37: Revised cement emissions in Uzbekistan.**

**Appendix E: Global Clinker Ratio**

The approximate implied global clinker ratio can be derived from emissions and cement production using default emission
factors (Figure E1). The trend up until 1990 is largely a result of the assumptions used in extrapolation, although in earlier
years the data for the US and Europe dominate.





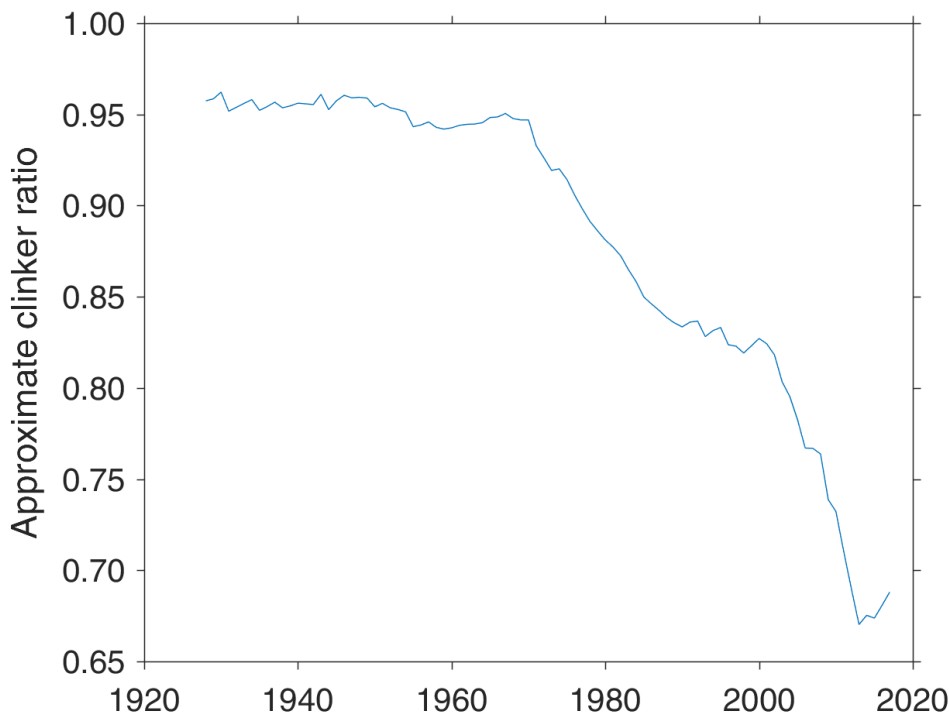

Figure E1: Implied global clinker ratio, derived from emissions estimates and cement production data.



**Competing Interests**

The authors declare that they have no conflict of interest.

**Acknowledgements**

Valuable assistance in this work was received from Hendrik van Oss of the US Geological Survey. Shaohui Zhang of IIASA
provided historical clinker and cement production data provided to him directly by the Chinese Cement Association. Chen Pan
of Nanjing University assisted with gathering additional data and information from Chinese sources. Gregg Marland of
Appalachian State University provided the report by Griffin. Funding was provided by CICEP – Strategic Challenges in
International Climate and Energy Policy (Research Council of Norway grant number 209701). This work builds on the
legacy of CDIAC, especially the work of Gregg Marland, Robert Andres, and Tom Boden.

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
