# Peer review of "Global CO2 emissions from cement production"

_Earth System Science Data, 2018_

## Referee Comment (RC1) · Anonymous Referee #1 · 24 Sep 2018

Overall a good and useful update from prior version, easy to read and access data.

A few small suggestions:

Living data process seems to work, changes clearly marked in this version of manuscript.

Data downloads easily, although I question significant figures. E.g. from .csv, USA in 1950 shows 19722.779928: 12 significant figures? Somehow I doubt that the author intended that precision.

The url link in the USGS 2014 reference does not work. Please check and update? The url link in the BP 2018 reference does not work. Please check and update? The url link in the UNFCCC 2018 reference does not work. Please check and update?

Based on these three quick checks, a careful check of the entire reference list to ensure

proper function of all urls seems in order?

Page 4 line 31: "IPCC Guidelines now recommend use of a default clinker ratio of 0.75" 0.75 should rather be expressed as 75%? But because of blending this number should be lower than prior estimates, e.g. lower than 60-67%. So the 0.75 represents a fraction of the default ratio, e.g. 0.75 times 60%, to give 45%? Author allows confusion here?

Page 5 line 27: here reader first encounters "tier" categories. Explain tiers 1, 2, 3?

Page 6 line 27: here reader finds a clinker ratio of 0.95, high as expected for OPC. But how does this number compare to the 0.75 earlier and to the clinker ratios of 60 to 67 % cited earlier. Confusing cement chemistry (64% CaO by weight) with clinker content? If this reader finds these numbers confusing, others will as well? Some hints evident at the top of page 8? Appendix A1 and likewise Appendix E supplies definitive information but we could at least have had a clearer summary and explanation of terms earlier? Perhaps a short table to define terms somewhere near the top of the Introduction section?

Page 24: I do not find reference to Figure D4 anywhere in the text?

Nice credit to CDIAC in the Acknowledgements.

---

## Referee Comment (RC2) · Anonymous Referee #2 · 4 Oct 2018

Dear editor,

Thank you for giving me the opportunity to review this manuscript.

This manuscript is an update to an important work that estimated global process CO2 emissions from cement production by including latest production statistics as well as revisiting assumptions on e.g. clinker-to-cement ratio values assumed in earlier studies.

While I enjoyed reading the manuscript and going through the datasets, it was not always clear the 'identify' of this manuscript provides compared to the earlier version https://doi.org/10.5194/essd-10-195-2018 as the two have identical titles. This manuscript also does not refer to the earlier article when a comparison of results is expected (in particular, cumulative emissions that changed substantially). Of course, this might be entirely my misunderstanding as I am not familiar with the practices of

the ESSD journal, the objectives of which are somewhat different from other academic journals.

Nevertheless, my assessment is that the manuscript needs revision before it can be accepted for publication. My comments are provided below:

General If I understood correctly this manuscript is an updated version of the article also published in ESSD https://doi.org/10.5194/essd-10-195-2018 – why does this current manuscript have exactly the same title as the already-published one? The conventions for ESSD might be different from those in other journals due to the focus of the journal, but from how the Global Carbon Budget updates are titled and that this manuscript covers one more data year (2017), shouldn't this manuscript have an original name if it is to be published as a new paper (and not a replacement of the aforementioned article)?

In relation to the above, I found it very strange that this manuscript does not cite the earlier article (essd-10-195-2018) at all. Even if this manuscript is an annual data update, it is recommended that the author clearly communicates the main changes on the methodology and the results (and main reasons for the changes).

The track change version (supplementary file) compared to the earlier article was very useful. Nevertheless, to make this manuscript more transparent without a track change file, it would be nice to have the main changes (or their summary) presented in a tabular format, e.g. two columns with 'before' and 'after'.

This manuscript presents both cumulative emissions and annual emissions. To avoid confusion, it is advisable to add "/year" for annual emission figures throughout the manuscript. A dataset that is currently not uploaded but would be very useful for the energy and climate policy/technology researchers would be country-level clinker ratio time series. Is possible to have this dataset on public domain?

On the references to Appendices: not all Appendices are referenced in the main text,

and the ones that are referenced do not appear in an alphabetical order (first Appendix A, then Appendix D, followed by Appendix C). Please re-organise.

Abstract "The required data for estimating emissions from global cement production are poor, and [...]" : it was not fully clear whether the data availability was poor or data quality was poor (or something else). Please clarify. "Cumulative emissions from 1928 to 2017 were 36.9±2.3 Gt CO2, 70% of which have occurred since 1990. Emissions in 2016 were 28% lower than those recently reported by the Global Carbon Project." Where in the main text is this substantiated?

Main text P4 l4: "Anonymous 2010" : Is the author PBL (as an organization) ? P6 l9: On "Process emissions from cement production reached a new peak in 2017 of 1.48±0.20 GtCO2", does the author mean a new historical high? When looking at the data file, 2017 emissions are 1.477095 GtCO2 whereas 2014 emissions are still a tiny bit higher at 1.478259 GtCO2. Please clarify. P6 l12: "Cumulative emissions over 1928–2017 were 36.9±2.3 GtCO2" – The value has reduced significantly from the previous version even after including 2017 data (39.3 GtCO2 for 1928-2016), but there is no clear explanation on what caused this change. Please elaborate. P6l 15: Since the China paragraph comes right after the global results, readers would expect to learn whether China is the main contributor to the 'new peak' in 2017 but this is unfortunately not entirely clear. It would be good to elaborate. P6 l27: "Uncertainty jumps" does not sound like a scientific expression. Suggest rewording. P10 l2: "It is intended that the database will be used in the Global Carbon Budget and updated annually, with both data updates and methodological improvements." – the author maintains the same sentence as in the earlier article (essd-10-195-2018), but the repetition of this makes the readers wonder why the dataset is still not used in the Global Carbon Budget project. If there are particular challenges or limitations regarding the dataset's inclusion in the Global Carbon Budget, please clearly describe them. P10 (conclusions): it would be important to mention that this is a 'living document/data' and will be updated annually.

---

## Author Comment (AC1) · 16 Oct 2018

**Responses to referees' comments**

Firstly I must say that the comments by the two reviewers are greatly appreciated. I'm certainly grateful to have reviewers that read through the article so closely. The manuscript has been improved in response to these reviews.

**Anonymous Referee #1**

Overall a good and useful update from prior version, easy to read and access data.
A few small suggestions:
Living data process seems to work, changes clearly marked in this version of manuscript.
Data downloads easily, although I question significant figures. E.g. from .csv, USA in 1950 shows 19722.779928: 12 significant figures? Somehow I doubt that the author intended that precision.

> Thank you for raising this. No, this certainly wasn't intentional. I have reduced the precision to four significant figures. I will upload the revised data files to the repository once the review process is complete.

The url link in the USGS 2014 reference does not work. Please check and update? The url link in the BP 2018 reference does not work. Please check and update? The url link in the UNFCCC 2018 reference does not work. Please check and update?
Based on these three quick checks, a careful check of the entire reference list to ensure proper function of all urls seems in order?

> Yes, agreed. I have rechecked all links and have updated the following that did not work:
> Ciais 2013; Korean Ministry of Environment 2012; MSTI 2016; WBCSD&IEA 2013
> Several links also required http:// in front for the PDF reader to open them correctly, and these are now fixed.
> Strangely, the BP link does work for me. I would hope that most readers, on finding that a link fails, would turn instead to an internet search engine, given that there is more than sufficient information.

Page 4 line 31: "IPCC Guidelines now recommend use of a default clinker ratio of 0.75" 0.75 should rather be expressed as 75%? But because of blending this number should be lower than prior estimates, e.g. lower than 60-67%. So the 0.75 represents a fraction of the default ratio, e.g. 0.75 times 60%, to give 45%? Author allows confusion here?

> I have chosen to use a decimal to represent the clinker ratio, but I believe this is a question of style.
> My statement here isn't actually that the IPCC default value is incorrect, rather I'm simply stating that they recommended this value in a particular situation. In an effort to clarify, I have reworded as:
> "the IPCC Guidelines now recommend inventory compilers use of a default clinker ratio of 0.75 when it is known that significant amounts of blended cements are produced in their country"
> So no, no further multiplication, simply 0.75 (75%).

Page 5 line 27: here reader first encounters "tier" categories. Explain tiers 1, 2, 3?

> I have added in parentheses immediately afterwards:

"the middle level of detail; Hanle et al., 2006"
Hanle et al is the Minerals chapter of the 2006 IPCC Guidelines where this is explained.

Page 6 line 27: here reader finds a clinker ratio of 0.95, high as expected for OPC. But how does this number compare to the 0.75 earlier and to the clinker ratios of 60 to 67 % cited earlier. Confusing cement chemistry (64% CaO by weight) with clinker content? If this reader finds these numbers confusing, others will as well? Some hints evident at the top of page 8? Appendix A1 and likewise Appendix E supplies definitive information but we could at least have had a clearer summary and explanation of terms earlier? Perhaps a short table to define terms somewhere near the top of the Introduction section?

Thank you for pointing out this structural problem. I have attempted to clarify by adding the following text to the second paragraph of the Introduction section:
"These emissions ($E$) can be calculated as:

$$E = \frac{M_r^{CO_2}}{M_r^{CaO}} f_{clink}^{CaO} f_{cem}^{clink} M_{cem}$$

where $M_r^{CO_2}$ is the molecular weight of $CO_2$, $M_r^{CaO}$ the molecular weight of CaO, $f_{clink}^{CaO}$ the fraction of CaO in clinker, $f_{cem}^{clink}$ the fraction of clinker in cement (the 'clinker ratio'), and $M_{cem}$ the mass of cement (see Appendix A for details)."

Page 24: I do not find reference to Figure D4 anywhere in the text?

A reference has now been added, although this is now C4 as the two appendices have been switched.

Nice credit to CDIAC in the Acknowledgements.

It was certainly appropriate, and should also have been there in the first version of the article.

**Anonymous Referee #2**

Dear editor,
Thank you for giving me the opportunity to review this manuscript.
This manuscript is an update to an important work that estimated global process CO2 emissions from cement production by including latest production statistics as well as revisiting assumptions on e.g. clinker-to-cement ratio values assumed in earlier studies.

While I enjoyed reading the manuscript and going through the datasets, it was not always clear the 'identify' of this manuscript provides compared to the earlier version https://doi.org/10.5194/essd-10-195-2018 as the two have identical titles. This manuscript also does not refer to the earlier article when a comparison of results is expected (in particular, cumulative emissions that changed substantially). Of course, this might be entirely my misunderstanding as I am not familiar with the practices of the ESSD journal, the objectives of which are somewhat different from other academic journals.

I must say this was also unclear to me, but following the reviewer's suggestion I have modified the title and referred to the previous version (see a later response).

Nevertheless, my assessment is that the manuscript needs revision before it can be accepted for publication. My comments are provided below:

General If I understood correctly this manuscript is an updated version of the article also published in ESSD https://doi.org/10.5194/essd-10-195-2018 – why does this current manuscript have exactly the same title as the already-published one? The conventions for ESSD might be different from those in other journals due to the focus of the journal, but from how the Global Carbon Budget updates are titled and that this manuscript covers one more data year (2017), shouldn't this manuscript have an original name if it is to be published as a new paper (and not a replacement of the aforementioned article)?

Thank you for picking up on this omission. I have now modified the title of the paper to "Global $CO_2$ emissions from cement production, 1928–2017", which will distinguish it from the previous version and any future versions.

In relation to the above, I found it very strange that this manuscript does not cite the earlier article (essd-10-195-2018) at all. Even if this manuscript is an annual data update, it is recommended that the author clearly communicates the main changes on the methodology and the results (and main reasons for the changes).

I have now referred to the previous version of the article in a new paragraph at the end of the Methods section (see next response).

The track change version (supplementary file) compared to the earlier article was very useful. Nevertheless, to make this manuscript more transparent without a track change file, it would be nice to have the main changes (or their summary) presented in a tabular format, e.g. two columns with 'before' and 'after'.

I have added a paragraph at the end of the Methods section that reads:
"This is the second version of this article in the living data format, updated from (Andrew, 2018). The main changes compared to that version are: (1) Removed double-counting of former Soviet countries in the global total; (2) Changed units of global emissions to match country-level emissions; (3) Updated to 2018 UNFCCC submissions by Annex-I countries; (4) Updated South Korea and Jamaica; (5) Added new estimates for Saudi Arabia, Togo, Israel, Lebanon, Serbia, Bosnia and Herzegovina, Morocco, Mongolia, and Colombia; (6) Added earlier estimates for Sweden; (7) Removed data for countries that no longer exist and non-countries (Antarctica and Kuwaiti Oil Fires)."
This information has also been add to a 'release_notes.txt' file that will be included in the next submission to the data repository.
Further, I have added the following text to the Results section:
"The main reason for the difference in the cumulative results presented here with those presented in the previous version of this "living data" article is the correction of double counting of the countries of the former Soviet Union."

This manuscript presents both cumulative emissions and annual emissions. To avoid confusion, it is advisable to add "/year" for annual emission figures throughout the manuscript.

I understand that this is common in some communities, but rather than clutter the manuscript now and potentially introduce an error by carelessly missing one or more

that should have '/year', I have instead added the following text in parentheses after the first result:
"unless clearly indicated, all emissions are reported as per year"
Whenever I have presented cumulative results, this is already clearly indicated.

A dataset that is currently not uploaded but would be very useful for the energy and climate policy/technology researchers would be country-level clinker ratio time series. Is possible to have this dataset on public domain?
I have been reluctant to publish these data in full since many of them are based on strong assumptions, and marking them up appropriately (which would be very time consuming) would not prevent people from misusing them as if they were primary data. The data files associated with the paper do include all original data, including clinker production and cement production data where available. I will however consider assembling the 'primary' production data (i.e. those officially supplied rather than based on my assumptions) into a single file, and may add that to the data repository at a later date. I am in strong agreement with the idea of making data publicly accessible.

On the references to Appendices: not all Appendices are referenced in the main text, and the ones that are referenced do not appear in an alphabetical order (first Appendix A, then Appendix D, followed by Appendix C). Please re-organise.
I have moved the text and figure from Appendix E to within Appendix A (Figure E1 becomes Figure A1, with a longer caption).
I have added a reference to Appendix B in the first paragraph of the Methods section.
The order of Appendix C and D has been reversed, and all subsection and figure labels have been changed accordingly.

Abstract "The required data for estimating emissions from global cement production are poor, and […]" : it was not fully clear whether the data availability was poor or data quality was poor (or something else). Please clarify.
Yes, this was poorly written. I have reworded as:
"The availability of the required data for estimating emissions from global cement production is poor…"

"Cumulative emissions from 1928 to 2017 were 36.9_2.3 Gt CO2, 70% of which have occurred since 1990. Emissions in 2016 were 28% lower than those recently reported by the Global Carbon Project." Where in the main text is this substantiated?
I have added some text to the first paragraph of the Results section to address this:
"In 2016 emissions were 1.46 Gt $CO_2$, 28% lower than the 2.02 Gt $CO_2$ obtained by the GCP by extrapolating CDIAC's estimates using global cement production data (Le Quéré et al., 2018)…. Cumulative emissions from 1928 to 2017 were 36.9±2.3 Gt $CO_2$, 70% of which have occurred since 1990."

Main text P4 l4: "Anonymous 2010" : Is the author PBL (as an organization) ?
Spurred on by the reviewer's question, I have confirmed the author of this piece and changed the citation accordingly. (It was Greet Janssens-Maenhoot.)

P6 l9: On "Process emissions from cement production reached a new peak in 2017 of 1.48_0.20 GtCO2", does the author mean a new historical high? When looking at the data file, 2017 emissions are 1.477095 GtCO2 whereas 2014 emissions are still a tiny bit higher at 1.478259 GtCO2. Please clarify.

> I can only assume that this is a result of a change in the data prior to submission, but even if the 2017 emissions were slightly higher than those in 2014 it is not appropriate to make this comment given the level of uncertainty. I have reworded as:
>
> "Process emissions from cement production reached 1.48±0.20 GtCO2 in 2017, returning to about the same level as in 2014 after having declined in the interim"

P6 l12: "Cumulative emissions over 1928–2017 were 36.9_2.3 GtCO2" – The value has reduced significantly from the previous version even after including 2017 data (39.3 GtCO2 for 1928-2016), but there is no clear explanation on what caused this change. Please elaborate.

> As noted above, I have added the following sentence at the end of the Results section:
>
> "The main reason for the difference in the cumulative results presented here with those presented in the previous version of this "living data" article is the correction of double counting of the countries of the former Soviet Union."

P6l 15: Since the China paragraph comes right after the global results, readers would expect to learn whether China is the main contributor to the 'new peak' in 2017 but this is unfortunately not entirely clear. It would be good to elaborate.

> I have added the following sentence to the China paragraph:
>
> "The rebound in Chinese cement production, and therefore emissions, is the main reason for global emissions to have regained the level of 2014."

P6 l27: "Uncertainty jumps" does not sound like a scientific expression. Suggest rewording.

> Agreed. I have replaced 'jumps' to 'increases sharply'.

P10 l2: "It is intended that the database will be used in the Global Carbon Budget and updated annually, with both data updates and methodological improvements." – the author maintains the same sentence as in the earlier article (essd-10-195-2018), but the repetition of this makes the readers wonder why the dataset is still not used in the Global Carbon Budget project. If there are particular challenges or limitations regarding the dataset's inclusion in the Global Carbon Budget, please clearly describe them.

> At the time of submission of the present manuscript the decision had not yet been formally made to use this dataset in the 2018 Global Carbon Budget. I have now modified the text as:
>
> "The database is used in the Global Carbon Budget for the first time in the 2018 edition, and the intention is that it will be updated annually, with both data updates and methodological improvements under the "living data" format."
>
> I can record here that the reason for the delay in deciding stemmed from consideration of the consequences of a shift from unintentionally including some

other carbonate emissions (by using CDIAC's estimates) to intentionally excluding them.

P10 (conclusions): it would be important to mention that this is a 'living document/data' and will be updated annually.

Indeed, see previous response.